

# Impact of particle number and mass size distributions of major chemical components on particle mass scattering efficiency in urban Guangzhou of South China

Jun Tao[1,*], Zhisheng Zhang[1], Yunfei Wu[2], Leiming Zhang[3,*], Zhijun Wu[4], Peng Cheng[5], Mei Li[5], Laiguo Chen[1], Renjian Zhang[2], Junji Cao[6]

[1]South China Institute of Environmental Sciences, Ministry of Environmental Protection, Guangzhou, China

[2]RCE-TEA, Institute of Atmospheric Physics, Chinese Academy of Sciences, Beijing, China

[3]Air Quality Research Division, Science and Technology Branch, Environment and Climate Change Canada, Toronto, Canada

[4]State Key Joint Laboratory of Environmental Simulation and Pollution Control, College of Environmental Sciences and Engineering, Peking University, Beijing, China

[5]Institute of Mass Spectrometer and Atmospheric Environment, Jinan University, Guangzhou, China

[6]Key Laboratory of Aerosol Chemistry and Physics, Institute of Earth Environment, Chinese Academy of Sciences, Xi'an, China

*Correspondence to: (Leiming Zhang) leiming.zhang@canada.ca or (Jun Tao) taojun@scies.org





**Abstract.** To grasp the key factors affecting particle mass scattering efficiency
(MSE), particle mass and number size distribution, bulk $PM_{2.5}$ and $PM_{10}$ and their
major chemical compositions, and particle scattering coefficient ($b_{sp}$) under dry
condition were measured at an urban site in Guangzhou, south China during
2015-2016. On annual average, $10\pm2\%$, $48\pm7\%$ and $42\pm8\%$ of $PM_{10}$ mass were in the
condensation, droplet and coarse modes, with mass median aerodynamic diameters
(MMADs) of $0.21\pm0.00$, $0.78\pm0.07$ and $4.57\pm0.42$ μm, respectively. The identified
chemical species mass concentrations can explain $79\pm3\%$, $82\pm6\%$ and $57\pm6\%$ of the
total particle mass in the condensation, droplet and coarse mode, respectively. Organic
matter (OM) and elemental carbon (EC) in the condensation mode, OM, $(NH_4)_2SO_4$,
$NH_4NO_3$ and crustal element oxides in the droplet mode, and crustal element oxides,
OM and $CaSO_4$ in the coarse mode were the dominant chemical species in their
respective modes. The measured $b_{sp}$ can be reconstructed to the level of $91\pm10\%$
using Mie theory with input of the estimated chemically-resolved number
concentrations of $NaCl$, $NaNO_3$, $Na_2SO_4$, $NH_4NO_3$, $(NH_4)_2SO_4$, $K_2SO_4$, $CaSO_4$,
$Ca(NO_3)_2$, OM, EC, crustal element oxides and unidentified fraction. MSEs of bulk
particle and individual chemical species were underestimated by less than 13 % in any
season based on the estimated $b_{sp}$ and chemical species mass concentrations. Seasonal
average MSEs varied in a small range of $3.5\pm0.1$ to $3.9\pm0.2$ $m^2$ $g^{-1}$ for fine particles,
which was mainly caused by seasonal variations of the mass fractions and MSEs of
OM in the droplet mode.

22          Keywords: particle size distribution, particle chemical composition, particle mass

scattering efficiency





## 1. Introduction

Light extinct coefficient ($b_{ext}$) of atmospheric particles, which is the sum of their
scattering ($b_{sp}$) and absorption ($b_{ap}$) coefficients, is a key index of haze weather (Hand
and Malm, 2007). In most cases, $b_{sp}$ accounted for more than 90% of $b_{ext}$ (Takemura
et al., 2002; Tao et al., 2017a). Numerous studies have demonstrated that haze is
mainly caused by high concentrations of fine particles (PM$_{2.5}$, with aerodynamic
diameter smaller than 2.5 μm) (Hand and Malm, 2007; Huang et al., 2012; Malm et
al., 1994; Malm et al., 2000; Malm et al., 2003; Malm and Hand, 2007; Sisler and
Latimer, 1993; Sisler et al., 1996; Sisler and Malm, 2000; Wang et al., 2014b; Zhao et
al., 2013). Knowledge of the dominant chemical species in PM$_{2.5}$ (e.g. $(NH_4)_2SO_4$,
$NH_4NO_3$ and OM) and their contributions to $b_{sp}$ is crucial in making feasible policies
for alleviating haze (Watson, 2002).
Generally, $b_{sp}$ can be estimated in reasonable accuracy using Mie theory when
size distributions of dominant chemical species are known (Cheng et al., 2008; Cheng
et al., 2009; Gao et al., 2015; Malm et al., 2003; Watson et al., 2008). However,
routinely monitoring of the size distributions of all the dominant chemical
components is impractical. To evaluate haze in the national parks in U.S.A. under the
Regional Haze Rule, the original and revised IMPROVE formulas were developed for
reconstructing $b_{sp}$ based on the chemical species in PM$_{2.5}$ and coarse particle mass
concentrations monitored in the IMPROVE network (Pitchford et al., 2007; Watson,
2002). The MSEs of chemical species are the important parameters of the IMPROVE
formulas for building the relationships between chemical species and $b_{sp}$ (Hand and





Malm, 2007). The recommended MSEs of $(NH_4)_2SO_4$, $NH_4NO_3$, OM and fine soil
(estimated from crustal elements) in $PM_{2.5}$ were 3.0, 3.0, 4.0 and 1.0 $m^2$ $g^{-1}$,
respectively, in the original IMPROVE formula. However, MSEs of any particle
species vary with its mass concentrations and size distributions (Lowenthal and
Kumar, 2004; Malm et al., 2003; Malm and Hand, 2007; Malm and Pitchford, 1997).
Subsequently, the MSEs and mass concentrations of $(NH_4)_2SO_4$, $NH_4NO_3$ and OM in
the $PM_{2.5}$ were separated into small and large modes in the revised IMPROVE
formula (Hand and Malm, 2007).
China has been suffering from severe $PM_{2.5}$ pollution and haze weather (Li et al.,
2016; Ming et al., 2017; Wang et al., 2017; Zhang et al., 2013). To investigate the
formation of haze, the original and revised IMPROVE formulas have been directly
applied in many cities in China (Hua et al., 2015; Shen et al., 2014; Tao et al., 2009;
Zhang et al., 2012a; Zou et al., 2018). The IMPROVE formulas have been proved to
over- or underestimate $b_{sp}$ in urban cities in China (Cao et al., 2012; Cheng et al.,
2015; Han et al., 2014; Jung et al., 2009a; Jung et al., 2009b; Tao et al., 2012; Tao et
al., 2014b), which were likely due to the significantly different size distributions of
the major chemical components and related mass fractions in $PM_{2.5}$ between different
countries or even cities (Bian et al., 2014; Cabada et al., 2004; Chen et al., 2017; Guo
et al., 2009; Lan et al., 2011; Tian et al., 2014b; Yao et al., 2003; Yu et al., 2010;
Zhang et al., 2008; Zhuang et al., 1999b). To reduce the uncertainties in the estimated
$b_{sp}$ using the original and revised IMPROVE formulas, the average MSEs of
dominant chemical species were typically estimated by the multiple linear regression



method (Hand and Malm, 2007). Although the estimated $b_{sp}$ by the multiple linear
regression model may be close to the measured $b_{sp}$, the rationality of the estimated
MSEs of chemical species were unknown (Tao et al., 2014a; Tao et al., 2014b; Tao et
al., 2015; Tao et al., 2016; Yao et al., 2010; Wang et al., 2014a).

According to Mie theory, variations in size distributions (e.g. MMADs and mass

fractions) of chemical components are the most important factors for hindering the
applications of the IMPROVE formulas and multiple linear regression models.
Although many studies have conducted on understanding size distributions and
chemical compositions of fine particles in China, few studies have explored the
relationship between the size distribution of major chemical species and their MSEs
(Cheng et al., 2008; Cheng et al., 2009; Gao et al., 2015). To fill this knowledge gap,
size-segregated particle mass, $PM_{10}$, $PM_{2.5}$ and their major chemical components, and
online data including size distribution of particle number, $b_{sp}$ under dry conditions and
water-soluble inorganic ions were synchronously measured at an urban site in
Guangzhou covering four seasons in 2015-2016. Size distributions of dominant
chemical components were first characterized in section 3.1, followed by discussions
on the closures of particle mass and number concentration and $b_{sp}$ in 3.2. Key factors
controlling the variations of chemical species and their MSEs were then discussed in
section 3.3. Knowledge gained from the present study will improve the assessments of
air-quality and climate impact caused by atmospheric particles.
**2. Methodology**
**2.1 Site description**

The observational site in urban Guangzhou is situated inside the South China



Institute of Environmental Science (SCIES) (23°07′N, 113°21′E) (Fig. 1) with no
obvious surrounding industrial activities. The instruments used in this study were
installed on the roof of a building 50 m above ground (Tao et al., 2018). The working
conditions of all the instruments were controlled under 26 degree in temperature and
40% in relative humidity by three air conditioners.

*Insert Figure 1*

**2.2 Field sampling**
Size-segregated particle samples were collected using Anderson 8-stage air
samplers with the cut-off points of 0.43, 0.65, 1.1, 2.1, 3.3, 4.7, 5.8 and 9.0 μm
(Thermo-electronic Company, USA). Two sets of samplers were used alternatively
due to the need of daily clearance of the instruments. The samplers were operated at
an airflow rate of 28.3 L min$^{-1}$. The sampling flow rate was controlled by a flow meter
(Aalborg Inc., USA). Samples were collected on 81 mm quartz fiber filter (Whatman
QM-A). Samples were collected during different seasons: 15 July- 6 August, 2015
(representative of summer), 15 October- 5 November, 2015 (autumn), 4-20 January,
2016 and 19-22 February, 2016 (winter), and 8-20 April, 2016 and 4-14 May, 2016
(spring). Sampling duration was 48 h in spring and 24 h in the other seasons, all
starting at 10:00 local time.
Bulk $PM_{2.5}$ and $PM_{10}$ samples were collected using two Gravisol Sequential
Ambient Particulate Monitor (GSAPM) samplers (APM Inc., Korea) at a flow rate of
16.7 L min$^{-1}$. Samples were collected on 47 mm quartz fiber filter (Whatman QM-A).
Sampling durations were the same as those for collecting size-segregated samples in
every season. The sampling information is summarized in Table 1. Moreover, 8 sets of





blank samples were also collected for each of the size-segregated particle, $PM_{2.5}$ and
$PM_{10}$ samples during the whole sampling period. The aerosol-loaded filter samples
were stored in a freezer at -18 °C before analysis to prevent volatilization of particles.

*Insert Table 1*

The background water-soluble inorganic ions (WSII) (e.g. $Na^+$, $Ca^{2+}$) of quartz
fiber filter were slightly high in general. Thus, 47mm and 81mm quartz fiber filters
were first baked at 500 °C for 3 h to remove adsorbed organic vapors; they were then
soaked in distilled-deionized water for 3 h for several times to remove WSII until the
background values were less than 0.01 mg $L^{-1}$. Finally, the quartz fiber filters were
dried through baking at 200 °C. All blank quartz fiber filters were stored in
desiccators.
Particle number concentration for particles in the range of 14 nm - 615 nm in
mobility diameter (default as geometric diameter ($D_g$)) was measured using a
scanning mobility particle sizer (SMPS; TSI Model 3936, TSI, Inc., St. Paul, MN)
combined with a long differential mobility analyzer (DMA; TSI Model 3080) and a
condensation particle counter (CPC; TSI Model 3010), and for particles in the range
of 542 nm - 10 μm aerodynamic diameter ($D_a$) using an Aerodynamics Particle Sizer
(APS; TSI Model 3321), both at 5 min resolution. Dry $b_{sp}$ was measured using a
single wavelength integrating nephelometer (Ecotech Pty Ltd, Australia, Model
Aurora1000G) at the wavelength of 520 nm. Ambient air passed through three total
suspended particulate (TSP) cyclones, then stainless steel tubes and the Nafion driers
prior to sampling by the SMPS, APS and nephelometer. RH of aerosol samples was
controlled to be lower than 30% by sweeping dry air from a compressed air pump.



Water-soluble inorganic ion ($NO_3^-$) was measured using an In-situ instrument of Gas
and Aerosol Composition (IGAC, Model S-611, Machine Shop, Fortelice
International Co., Ltd., Taiwan, China) at a resolution of 1-h (Tao et al., 2018).
**2.3 Lab chemical analysis and data quality assurance and control**

47 mm and 81 mm quartz fiber filters were measured gravimetrically for particle

mass concentration using a Sartorius ME 5-F electronic microbalance with a
sensitivity of ±1 μg (Sartorius, Göttingen, Germany) after 24 h equilibration at
temperature of 23±1 °C and RH of 40±5%. Microbalance was calibrated by 5 mg, 200
mg and 5000 mg weights before weighting. Each filter was weighed at least three
times before and after sampling. Differences among replicate weights were mostly
less than 20 μg for each sample. Net mass was obtained by subtracting pre-weight
from post-weight.

Three pieces of 0.526 $cm^2$ punches from each 47 mm quartz filter samples and

one-fourth of each 81 mm quartz filter samples were used to determine water-soluble
inorganic ions. The extraction of water-soluble species from each filter was put into a
separate 4 mL bottle, followed by 4 mL distilled-deionized water (with a resistivity
of >18 MΩ), and then subjected to ultrasonic agitation for 1 h for complete extraction
of the ionic compounds. The extract solutions were filtered (0.25 μm, PTFE,
Whatman, USA) and stored at 4 °C in pre-cleaned tubes until analysis. Cation ($Na^+$,
$NH_4^+$, $K^+$, $Mg^{2+}$ and $Ca^{2+}$) concentrations were determined by ion chromatography
(Dionex ICS-1600) using a CS12A column with 20 mM Methanesulfonic Acid eluent.
Anions ($SO_4^{2-}$, $NO_3^-$, $Cl^-$, and $F^-$) were separated on an AS19 column in ion
chromatography (Dionex ICS-2100), using 20 mM KOH as the eluent. A calibration
was performed for each analytical sequence. Procedural blank values were subtracted

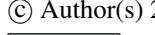



from sample concentrations. Method detection limits (MDL) of ions were within the
range of 0.001 to 0.002 mg L$^{-1}$.

OC and EC were analyzed using a DRI model 2001 carbon analyzer (Atmoslytic,

Inc., Calabasas, CA, USA). An area of 0.526 cm$^2$ punched from each 47mm quartz
filter and 1-4 dots punched from each 81mm quartz filter were analyzed for four OC
fractions (OC1, OC2, OC3, and OC4 at 140 °C, 280 °C, 480 °C, and 580 °C,
respectively, in a helium [He] atmosphere); OP (a pyrolyzed carbon fraction
determined when transmitted laser light attained its original intensity after oxygen [O$_2$]
was added to the analyzed atmosphere); and three EC fractions (EC1, EC2, and EC3
at 580 °C, 740 °C, and 840 °C, respectively, in a 2% O$_2$/98% He atmosphere). Here,
OC is operationally defined as OC1 + OC2 + OC3 + OC4 + OP and EC is defined as
EC1 + EC2 + EC3 – OP for 47mm samples. However, OC is operationally defined as
OC1 + OC2 + OC3 + OC4 and EC is defined as EC1 + EC2 + EC3 for 81mm
samples due to extremely low OP level. Average field blanks were subtracted from
each sample filter. MDLs of OC and EC were 0.41±0.2 µgC cm$^{-2}$ and 0.03±0.2 µgC
cm$^{-2}$, respectively.

To obtain high quality data of the size distributions of major chemical

components, bulk PM$_{2.5}$ and PM$_{10}$ samples were synchronously collected and the
same chemical components were analyzed. Generally, good correlations ($R^2$>0.90)
were found in the mass concentrations of the total particle and major chemical
components (including total carbon (TC), NO$_3^-$ and SO$_4^{2-}$) between the
size-segregated samples (PM$_{10}$ and PM$_{2.1}$) and the GSAPM samplers (PM$_{10}$ and
PM$_{2.5}$). The regression slopes were in the range of 0.91- 1.05, suggesting good and



acceptable data quality of the size distributions of the major chemical components
(Fig.S1).
**2.4 Data analysis methods**

In this work, the cut-off point 2.1 μm was chosen to separate the fine and coarse

mode particles for investigating the impact of aerosol size distribution on their
respective MSEs. Moreover, the cut sizes of <0.43 μm and 0.43 - 2.1 μm were used to
separate the condensation mode and droplet mode, respectively. Continuous
size-distribution profiles of major chemical components are needed in order to
accurately calculate $b_{sp}$ using Mie theory, and are obtained from the inversion of the
measured mass concentration distribution in the size bins of the Anderson 8-stage air
samplers using the technique described by Dong et al. (2004). However, this approach
is not applicable for the condensation mode because there is only one size bin in this
mode. Thus, MMADs of this mode are used for generating the continuous size
distributions of all the concerned chemical species. MMADs of this mode are
calculated according to:

$D_p = (D_{p1} \times D_{p2})^{0.5}$                                    (1)

Where $D_{p1}$ and $D_{p2}$ represent the lower (0.10 μm, limits of detection of Anderson

8-stage air sampler) and upper (0.43 μm) boundaries of this size bin, respectively. To
improve the resolution of $b_{sp}$, 401 bins were used for chemical species ranging from
10 nm to 100 μm, with a constant ratio between the adjacent size bins, defined as
$\log(D_{p2}/D_{p1}) = 0.01$. Further increasing the number of size bins does not have any
significant impact on the results, e.g., the changes in $b_{sp}$ are smaller than 1% even if
the above ratio of 0.01 is replaced with 0.001.

The ISORROPIA II model was run at the reserved mode (Fountoukis and Nenes,

2007) with input data of $K^+$, $Ca^{2+}$, $Mg^{2+}$, $NH_4^+$, $Na^+$, $SO_4^{2-}$, $NO_3^-$, $Cl^-$, RH (40%), and



temperature (25°C), to estimate the size-resolved mass concentrations of NaCl,
$NaNO_3$, $Na_2SO_4$, $NH_4Cl$, $NH_4NO_3$, $(NH_4)_2SO_4$, KCl, $KNO_3$, $K_2SO_4$, $MgCl_2$,
$Mg(NO_3)_2$, $MgSO_4$, $CaCl_2$, $Ca(NO_3)_2$, $CaSO_4$ and $H_2O$.
**3. Results and Discussion**
**3.1 Size distributions of total particle mass and major chemical components**
**3.1.1 Total particle mass**
Generally, any particle size distribution can be fitted into a combination of
condensation, droplet and coarse modes (John et al., 1990). Continuous log-normal
size distributions of particle mass including the condensation, droplet and coarse
modes were calculated using the method described in section 2.4 and are summarized
in Table 2. On annual average, $10\pm2\%$, $48\pm7\%$ and $42\pm8\%$ of $PM_{10}$ mass were in the
condensation, droplet and coarse modes, with the average MMADs of $0.21\pm0.00$,
$0.78\pm0.07$ and $4.57\pm0.42$, respectively. This result was comparable with those
observed by the Micro-Orifice Uniform Deposit Impactor (MOUDI) in other cities
(e.g. Shenzhen and Hong Kong) of the PRD region (Bian et al., 2014; Lan et al., 2011;
Yu et al., 2010).
The estimated annual $PM_{2.5}$ concentration based on the continuous log-normal
size distribution was $36.4\pm13.2$ μg m$^{-3}$, which was close to the synchronously
measured $PM_{2.5}$ ($36.8\pm15.3$ μg m$^{-3}$), although slightly higher than the sum of the mass
concentrations ($34.9\pm13.8$ μg m$^{-3}$) in the condensation and droplet modes. Thus, the
fine (sum of condensation and droplet) mode particles can reasonably represent $PM_{2.5}$.
Seasonal average particle mass concentrations were evidently lower in summer than
in the other seasons for all the three modes, were close during the other seasons for
the condensation and droplet modes, and were slightly higher in autumn and spring
than winter for the coarse mode. These results agree with the seasonal variations of





PM$_{2.5}$ observed at the same site in 2009-2010 (Tao et al., 2014b).

*Insert Table 2*


**3.1.2 Water-soluble inorganic ions**

Generally, SO$_4^{2-}$, NO$_3^-$ and NH$_4^+$ are the dominant WSIIs, especially in the

condensation and droplet modes. They are mainly formed through aqueous-phase
reactions in moisture conditions in the PRD region (Lan et al., 2011; Yu et al., 2010).
As expected, 77±6% SO$_4^{2-}$, 46±16% NO$_3^-$ and 89±7% NH$_4^+$ mass concentrations were
in the droplet mode on annual average (Table 2). Much lower fractions for NO$_3^-$ than
SO$_4^{2-}$ and NH$_4^+$ in the droplet mode were mostly due to the high volatility of NH$_4$NO$_3$
(Zhang et al., 2008). The MMADs of the three ions in the droplet mode were in the
range of 0.70-0.94 µm, comparable with MOUDI measurements (0.78-1.03 µm)
conducted in the PRD region (Bian et al., 2014; Lan et al., 2011; Yu et al., 2010).

Small fractions of SO$_4^{2-}$, NO$_3^-$ and NH$_4^+$ masses were distributed in the

condensation mode, e.g., 12±4%, 10±4% and 6±5%, respectively, on annual average.
The mass fractions of SO$_4^{2-}$ in the condensation mode shown above were much lower
than those (24-29%) observed in urban Guangzhou in 2006-2007 (Yu et al., 2010),
suggesting gas-phase chemical reactions of SO$_2$ has become less important in the
formation of SO$_4^{2-}$, likely due to the dramatic reduction of SO$_2$ emissions in urban or
suburban Guangzhou in the recent decade (Zheng et al., 2009; Zheng et al., 2018).
Note that the MMAD in the condensation mode was 0.21 µm in Yu et al. (2010), a
value that is comparable with those (0.23-0.42 µm) measured using MOUDI at urban
sites in the PRD region (Bian et al., 2014; Lan et al., 2011; Yu et al., 2010).

11±5% SO$_4^{2-}$, 44±18% NO$_3^-$ and 5±4% NH$_4^+$ mass concentrations were

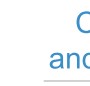
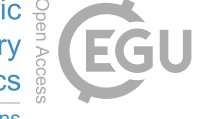

distributed in the coarse mode. In general, $NO_3^-$ mainly exists in the form of $NH_4NO_3$
in the condensation and droplet modes and associates with base cations in the coarse
mode (e.g., $Ca(NO_3)_2$ and $NaNO_3$) (Zhang et al., 2015a). More than 50% $NO_3^-$ mass
concentrations were distributed in the coarse mode in summer and autumn when
ambient temperatures were high. The MMADs of $NO_3^-$ in the coarse mode were
4.15±0.52 and 4.36±0.31 μm in summer and autumn, respectively, slightly lower than
those of $Ca^{2+}$ (4.10±0.42 and 4.72±0.47 μm in the same seasons), but evidently higher
than those of $Na^+$ (3.60±0.19 and 3.64±0.27 μm) (Table 2). This suggests that
$NH_4NO_3$ was prone to dissociate to $HNO_{3(g)}$ in summer and autumn due to the high
ambient temperatures with released $HNO_{3(g)}$ further reacting with mineral dust and to
a less extent with sea salt particles. In comparison, the MMADs of $SO_4^{2-}$ in the coarse
mode were in between of those of $Ca^{2+}$ and $Na^+$, likely due to uptake of $H_2SO_{4(g)}$ by
both mineral dust and sea salt particles (Zhang et al., 2015a). In contrast, the MMAD
of $NH_4^+$ in the coarse mode was 3.25±0.69 μm, much smaller than those of $SO_4^{2-}$ and
$NO_3^-$, suggesting that $NH_4^+$ in the coarse mode was likely from hygroscopic growth of
$NH_4^+$ in the droplet mode (Tian et al., 2014a).
It is also worth mentioning that most of $Cl^-$ was distributed in the coarse mode
and its MMAD (3.77±0.35 μm) was very close to that of $Na^+$ (3.75±0.38 μm),
especially in summer when air masses were originated from the China South Sea (Tao
et al., 2017b; Xia et al., 2017). The mole ratios of $Cl^-/Na^+$ were less than 1.0 in all the
seasons but spring due to the reactions between sea salt and acid gasses ($HNO_{3(g)}$ and
$H_2SO_{4(g)}$) (Zhuang et al., 1999a). The excess $Cl^-$ in the coarse mode in spring was
likely due to the aged biomass burning particles from the southeast Asian. In fact, the
concentration of the typical biomass burning tracer $K^+$ in the coarse mode was higher
in spring than in the other seasons (Zhang et al., 2015c). In any case, sea salt was



mainly distributed in the coarse mode rather than the droplet mode in urban
Guangzhou.
**3.1.3 OC and EC**

OC and EC in fine particles can be produced from both primary emissions of

vehicle exhaust, coal combustion, biomass burning and secondary formation (Chow et
al., 2011; Gentner et al., 2012; Gentner et al., 2017; Hallquist et al., 2009; Zheng et al.,
2006). In general, fresh OC and EC particles emitted from vehicle exhaust, coal
combustion and biomass burning should be distributed in the condensation mode
(Schwarz et al., 2008; Zhang et al., 2012b). Only 13±4% of OC and 31±7% of EC
mass concentrations were distributed in the condensation mode in the present study
(Table 2). OC/EC ratios were in the range of 0.9-1.6 in the condensation mode,
suggesting that vehicle exhaust was the dominant source of OC and EC in this particle
size range (Huang et al., 2006a; Schwarz et al., 2008; Shiraiwa et al., 2007; Watson et
al., 2001; Wu et al., 2017). 62±9% of OC and 55±7% of EC mass concentrations were
distributed in the droplet mode (Table 2), similar to that of $SO_4^{2-}$. These numbers were
similar to those observed in the other cities of the PRD region, and was previously
identified to be mainly caused by in-cloud aerosol processing (Huang et al., 2006b).
Cloud processing indeed plays important roles in forming droplet mode aerosols in
urban Guangzhou (Tao et al., 2018). OC/EC ratios were in the range of 2.2-3.2 in the
droplet mode, much higher than those in the condensation mode, suggesting that OC
in the droplet mode was mainly aged or secondary particles (Day et al., 2015; Huang
et al., 2006a; Wu and Yu, 2016).

Although only one size bin was available in the condensation mode in this study,

the estimated MMADs of OC and EC in this mode were comparable with those
(0.25-0.34 μm) measured using MOUDI with 3 bins in this size range at suburban





sites (e.g. Hong Kong and Shenzhen) (Lan et al., 2011; Yu et al., 2010). The MMADs
of OC and EC in the droplet mode were 0.76±0.07 µm and 0.66±0.08 µm,
respectively, which were slightly lower than those (0.7-1.0 µm for OC and 0.8-1.0 µm
for EC) found in earlier studies in the PRD region (e.g. Guangzhou, Hong Kong and
Shenzhen) (Lan et al., 2011; Yu et al., 2010). Noticeably, the MMADs of OC and EC
in the droplet mode were very close to those (0.73 µm for OC and 0.77 µm for EC)
measured in summer at a suburban site of Hong Kong, where the loadings of the
dominant chemical components (e.g. OC, EC and $SO_4^{2-}$) were low (Yu et al., 2010).

Road dust and biogenic aerosols were generally considered as the major sources

of OC and EC in the coarse mode (Ho et al., 2003; Zhang et al., 2015b). Significant
fractions of OC (25±8%) and EC (14±7%) mass concentrations were distributed in the
coarse mode. These numbers were comparable with those (13-38% for OC and 4-16%
for EC) measured at suburban sites of Guangzhou, Shenzhen and Hong Kong (Lan et
al., 2011; Yu et al., 2010), but were lower than those (51-57% for OC and 17-21% for
EC) measured in urban Guangzhou in 2006-2007. The MMADs of OC (3.73±0.58 µm)
and EC (3.69±0.65 µm) in the coarse mode were close to those (3.8-4.3 µm for OC
and 3.7-4.1 µm for EC) measured in suburban of Hong Kong, although smaller than
those (4.8-5.2 µm for OC and 5.0-5.2 µm for EC) measured in suburban of Shenzhen
and urban of Guangzhou (Lan et al., 2011; Yu et al., 2010). These results suggested
that the MMADs of OC and EC might decrease with their decreasing coarse mode
mass fractions. Annual $PM_{10}$ concentrations in 2015-2016 in this study were 40%
lower than those in 2006-2007 in the PRD region, which further supported the above
hypothesis (Yu et al., 2010).



### 3.2 Closures of particle mass and number concentrations and $b_{sp}$

### 3.2.1 Closure of particle mass concentration

To investigate the impact of chemical species in different size modes on $b_{sp}$, particle mass concentrations in the different modes were first reconstructed based on mass concentrations of individual known chemical components. The dominant water-soluble inorganic species including $NaCl$, $NaNO_3$, $Na_2SO_4$, $NH_4NO_3$, $(NH_4)_2SO_4$, $K_2SO_4$, $CaSO_4$ and $Ca(NO_3)_2$ were determined using the ISORROPIA II thermodynamic equilibrium model as mentioned in section 2.4. A ratio of OM to OC of 1.4, 1.6 and 1.6 would be appropriate for the condensation, droplet and coarse mode, respectively, which was based on the findings of a previous study that suggested an average OM/OC ratio of 1.57 and a range of 1.4-1.8 in an urban environment of the PRD region (He et al., 2011). In our previous study (Tao et al., 2017b), mass concentration of crustal element oxides in $PM_{2.5}$ was estimated from the measurements of five crustal elements (Al, Si, Ca, Fe and Ti) in urban Guangzhou. This approach cannot be used in the present study due to the lack of crustal elements measurements. Alternatively, crustal element oxides mass concentration was estimated from $Ca^{2+}$ mass concentration because of their good correlations as was found in our previous study (Fig. S2) (Tao et al., 2017b). Moreover, source profiles of soil dust (representing crustal element oxides) in cities of southern China also suggested that $Ca^{2+}$ accounted for 5% of total soil dust in $PM_{2.5}$ (Sun et al., 2019). On annual average, the estimated crustal element oxides accounted for 8±2%, 10±4% and 29±5% of the total particle mass concentrations in the condensation, droplet and coarse mode, respectively. The reconstructed mass concentrations accounted for 79±3%, 82±6% and 57±6% of the total in the condensation, droplet and coarse mode, respectively.



As shown in Fig. 2, OM, EC, $(NH_4)_2SO_4$, $NH_4NO_3$ and crustal element oxides
dominated in different modes in four seasons. For example, OM and EC accounted
for 31-39% and 14-19%, respectively, of particle mass in the condensation mode, OM,
$(NH_4)_2SO_4$, crustal element oxides and $NH_4NO_3$ accounted for 19-34%, 18-22%, 6-15%
and 4-11%, respectively, in the droplet mode, and crustal element oxides, OM and
$CaSO_4$ accounted for 22-34%, 12-17% and 4-5%, respectively, in the coarse mode. In
addition, the total of the other identified chemical species only accounted for less than
10% of the total particle mass in every mode. For example, $Na_2SO_4$ and $K_2SO_4$
mainly distributed in the droplet mode and together they accounted for only 2-5% of
the particle mass in this mode. NaCl, $NaNO_3$ and $Ca(NO_3)_2$ mainly distributed in the
coarse mode and each of these species accounted for less than 2% of the total particle
mass in this mode.

*Insert Figure 2*


**3.2.2 Closure of particle number concentration**
To estimate the contribution of individual chemical species on $b_{sp}$ using Mie
theory, number size distributions of the dominant chemical species were needed and
were calculated according to the method described in Lin et al. (2014). As shown in
Fig. 3, most chemical species (except $(NH_4)_2SO_4$ in summer) had much higher
number concentrations in the condensation than droplet or coarse mode. Although the
estimated number median aerodynamic diameters (NMADs) of the number
concentrations of individual chemical species mainly distributed in the range of
100-120 nm, the estimated NMADs of particle number concentrations were always
close to about 100 nm in four seasons. This was because the estimated MMADs of

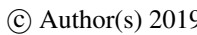



particle mass concentrations were a constant value (0.21 μm) based on only one bin in
the condensation mode. Moreover, average densities of particle in the condensation
mode only changed a little in four seasons because OM and EC were always the
dominant chemical species in this mode.
In contrast, the NMADs of particle number concentrations simultaneously
measured by the SMPS and APS distributed in the range of 40-80 nm (Fig. 4), which
shifted to smaller sizes than those estimated from the size-segregated chemical
species mass concentrations. This was because SMPS and APS collected dried
particles while the size-segregated sampler collected ambient particles. $D_g$ of particles
measured by SMPS can be converted to $D_a$ using the average particle density
calculated from the synchronously measured size-segregated individual chemical
species mass concentrations and densities. In any case, the NMADs of particle
number concentrations were less than 100 nm regardless of using SMPS and APS
measurements or the estimated size-segregated chemical species mass concentrations.
As shown in Fig. 3 and Fig. 4, most of particle numbers were in the range of 10 -
400 nm either observed by the SMPS or estimated from the size-segregated chemical
species mass concentrations. Total particle number concentration in the range of 10
nm-10 μm measured by the SMPS and APS were 7038±2250 cm$^{-3}$, 9774±1471 cm$^{-3}$,
5694±1942 cm$^{-3}$ and 10801±2986 cm$^{-3}$, respectively, in spring, summer, autumn and
winter, which were 1.09±0.24, 2.66±0.48, 1.05±0.20 and 2.33±0.67 times of those
estimated by the size-segregated chemical species mass concentrations.
NMADs estimated from the size-segregated chemical species mass
concentrations were close to those measured by the SMPS and APS in spring and
autumn, resulting in the close estimation of particle number concentrations to the
measured ones. In contrast, the estimated particle number concentrations from the the





size-segregated chemical species mass concentrations were evidently lower than those
measured by the SMPS and APS in summer and winter, due to the much higher
NMADs (100 nm) estimated from the size-segregated chemical species mass
concentrations than those (about 30 or 40 nm) measured by the SMPS and APS.
Fortunately, the single particle scattering efficiencies of chemical species in the
condensation mode at the wavelength 520 nm were much lower than those in the
droplet mode and the coarse mode (Fig S3).

To exclude the large uncertainties in the estimated particle number concentration

caused by condensation mode particles (which were due to the design flaws of
size-segregated sampler), particles smaller than 430 nm were not included in the
calculation below. The estimated particle number concentrations in the range of 430
nm-10 µm based on the size-segregated chemical species mass concentrations were
only slightly higher than those measured by the SMPS and APS. This was likely
because particles in the droplet mode may shift to the smaller sizes (<430 nm) during
the dry process by Nafion tube. Moreover, most of EC particles in the droplet mode
were internally mixed with OM or inorganic salts in the real world, which also may
result in overestimating the particle number concentrations by the size-segregated
chemical species mass concentrations (Wu et al., 2016; Yu et al., 2010). Correlation
coefficients between the estimated and measured particle number concentrations in
the range of 430 nm-10 µm were significantly improved when the intercepts in the
linear regression equations were retained. To some extent, the intercepts represented
the measurement or estimation errors of SMPS and APS and models. In any case,
good correlations ($R^2$>0.81) between the estimated daily particle number
concentrations and the measured ones were found and the slopes ranged from 0.79 to
1.03 in the four seasons (Fig. 5). These results suggested that the estimated particle



number concentrations were acceptable in the range of 430 nm-10 μm, noting that
particles in this size range dominate particle scattering efficiency.

*Insert Figure 3*

*Insert Figure 4*

*Insert Figure 5*



**3.2.3 Closure between the measured and estimated b$_{sp}$**
Although the number concentrations in the condensation mode were
underestimated, good correlations ($R^2 > 0.92$) were found between the measured and
estimated b$_{sp}$ with the slopes being 0.87, 0.87, 0.85 and 0.89 in spring, summer,
autumn and winter, respectively (Fig 6). On annual arithmetic average, the estimated
b$_{sp}$ can explain 91±10% of the measured b$_{sp}$. The residual fractions were likely related
to the chosen convert factor between OM and OC, measurements and sampling errors
of chemical species (especially NO$_3^-$), errors from the models (ISORROPIA II model,
Mie model, and especially the inversion technique method), and measurement errors
caused by the size-segregated sampler (Vaughan, 1989). Magnitudes of the
uncertainties caused by these sources are discussed below.
Although the convert factor of 1.6 between OM and OC was reasonable in urban
environment, a value of as high as 1.8 was found in literature (He et al., 2011). In
addition, OC mass concentrations were likely underestimated due to the OC/EC
protocol for size-segregated samples. Nevertheless, the estimated b$_{sp}$ can only be
increased by less than 3% if increasing the convert factor to 1.8 in the droplet mode.
Note that a previous study at the Fresno Supersite increased the estimated b$_{sp}$ by about



10% when increasing the convert factor from 1.4 to 1.8, likely due to the high mass
fraction of OC in fine particle at this site (Watson et al., 2008).
Different from the other chemical species, $NH_4NO_3$ can dissociate into $HNO_{3(g)}$
and $NH_{3(g)}$ during the filter gravimetric weighing process under dry condition. To
evaluate the evaporative loss of $NH_4NO_3$, synchronous online data of $NO_3^-$ were also
measured by an In-situ Gas and Aerosol Composition monitoring system at hourly
temporal resolution (Fig. S4). Seasonal average $NO_3^-$ concentrations were 42%
($PM_{2.5}$), 39% ($PM_{10}$), 42% ($PM_{2.5}$) and 19% ($PM_{2.5}$) less from filter measurements
than online measurements in spring, summer, autumn and winter, respectively.
Adjusting the filter $NO_3^-$ data using the above ratios can increase the estimated $b_{sp}$ by
7%, 2%, 4% and 2% in the respective season.

*Insert Figure 6*

Meanwhile, the measured $b_{sp}$ could also be underestimated due to the
dissociation of $NH_4NO_3$ during the dry processes of ambient particles through the
Nafion dryer. A previous study indicated the measured $b_{sp}$ being decreased by less
than 10% due to the dissociation of $NH_4NO_3$ in a heated nephelometer (Bergin et al.,
1997). In the present study, the chamber temperatures of nephelometer were less
than 300 K and the particle residence time in both the Nafion dryer and the
nephelometer chamber was about 7 seconds. Thus, the bias in the measured $b_{sp}$
should be less than 2% in any season according to the relationship among the loss
of $b_{sp}$, residence time and the temperature in chamber in a previous study (Bergin et
al., 1997). Combining all of the above-mentioned factors, the adjusted estimated $b_{sp}$
would increase to the level of 92%, 87%, 87% and 89% of the measured $b_{sp}$ in





spring, summer, autumn and winter, respectively. This means the above methods for
estimating $b_{sp}$ were reasonable with the adjusted estimated values explaining 87-92%
of the measured values after the filter-based $NO_3^-$ concentrations were adjusted
based on the online data. Thus, the errors from the models and size-segregated
samplers may account for remaining 8-13% of the measured $b_{sp}$.
Generally, the estimated seasonal average $b_{sp}$ were 146±40 $Mm^{-1}$, 99±33 $Mm^{-1}$,
169±54 $Mm^{-1}$ and 151±71 $Mm^{-1}$ in spring, summer, autumn and winter,
respectively (Fig. 7). The particles in the condensation, droplet and coarse modes
contributed 6-7%, 81-86% and 8-12%, respectively, to the estimated $b_{sp}$. OM and
EC were the dominant contributors, accounting for 32-41% and 30-37%,
respectively, of the estimated $b_{sp}$ in the condensation mode. OM and secondary
inorganic aerosols (sum of $(NH_4)_2SO_4$ and $NH_4NO_3$) were the dominant
contributors, accounting for 27-44% and 27-34%, respectively, of the estimated $b_{sp}$
in the droplet mode. Unidentified fraction, crustal element oxides and OM were the
dominant contributors, accounting for 26-47%, 16-29% and 19-27%, respectively,
of the estimated $b_{sp}$ in the coarse mode. The sum of the dominant contributors,
including OM, EC, secondary inorganic aerosols and crustal element oxides,
accounted for 70-79% of the estimated $b_{sp}$ in the four seasons. In contrast, the sum
of the other chemical species (including NaCl, $NaNO_3$, $Na_2SO_4$, $K_2SO_4$, $CaSO_4$,
$Ca(NO_3)_2$, $H_2O$) accounted for 5-10% and the unidentified fraction, 12-23% of the
estimated $b_{sp}$. In conclusion, visibility degradation was determined by the dominant
chemical species (e.g. OM, EC, secondary inorganic aerosols and crustal element
oxides) in the fine mode (both condensation and droplet), which agreed with the
results of the original and revised IMPORVE formulas (Pitchford et al., 2007).



*Insert Figure 7*


**3.3 Key factors for variations of particle and chemical species MSEs**
**3.3.1 The estimated MSEs of particle and chemical species**

To conveniently explore the control factors of particle MSE, the dominant

chemical species' MSEs were estimated by their mass concentrations and the
estimated $b_{sp}$, according to the measured chemical species mass concentrations in
section 3.1 and the estimated $b_{sp}$ in section 3.2. Here, only the MSEs of particle,
$(NH_4)_2SO_4$, $NH_4NO_3$, OM, EC, crustal element oxides and unidentified fraction in the
condensation, droplet, coarse, and fine modes (sum of condensation and droplet
modes) were estimated (Table 3), considering these chemical species accounted for
more than 90% of the estimated $b_{sp}$. Moreover, an external mixing of individual
chemical species was assumed in the estimation.

*Insert Table 3*


Undoubtedly, the particle MSE should be underestimated because the estimated

$b_{sp}$ was 11-15% less of the measured $b_{sp}$ in four seasons, as discussed in section 3.2.
The measured $b_{sp}$ would be biased low by about 3% due to the evaporation of
$NH_4NO_3$, while the $NO_3^-$ mass concentrations based the filter measurements were
biased low by 5%, 3%, 9% and 6% in spring, summer, autumn and winter,
respectively. Thus, the MSEs of $NO_3^-$ would be underestimated by 9%, 13%, 6% and
5% in the respective season in the real world. In conclusion, the MSEs of particle and
chemical species were underestimated by less than 13%.

On annual average, the estimated particle MSEs in the condensation, droplet and



coarse modes were 2.1±0.2 m$^2$ g$^{-1}$, 4.3±0.2 m$^2$ g$^{-1}$ and 0.5±0.0 m$^2$ g$^{-1}$, respectively.
The estimated particle MSE in the fine mode (sum of condensation and droplet modes,
similar to PM$_{2.5}$) was 3.7±0.2 m$^2$ g$^{-1}$, which was slightly higher than the value of 3.5
m$^2$ g$^{-1}$ estimated in 2009-2010 in urban Guangzhou (Tao et al., 2014b). Seasonal
variations of the estimated MSEs in the fine mode followed the sequence of winter
(3.9±0.2 m$^2$ g$^{-1}$) > autumn (3.8±0.2 m$^2$ g$^{-1}$) > summer (3.6±0.2 m$^2$ g$^{-1}$) > spring
(3.5±0.1 m$^2$ g$^{-1}$). Evidently, the estimated MSEs in the fine mode were slightly higher
in autumn and winter than spring and summer, which also agreed with the previous
studies in urban Guangzhou (Andreae et al., 2008; Jung et al., 2009a).
On annual average, the estimated MSEs of (NH$_4$)$_2$SO$_4$, NH$_4$NO$_3$, OM and crustal
element oxides (equal to fine soil in the IMPROVE formulas) in the fine mode were
4.4±0.8, 4.5±1.5, 4.6±0.3 and 2.6±0.1 m$^2$ g$^{-1}$, respectively, which were higher than
those (3.0, 3.0, 4.0 and 1.0 m$^2$ g$^{-1}$, respectively) from using the original IMPROVE
formula (Hand and Malm, 2007; Malm and Hand, 2007; Pitchford et al., 2007). As
shown in Table 3, the MSEs of (NH$_4$)$_2$SO$_4$, NH$_4$NO$_3$, OM and crustal element oxides
in the fine mode depended on their mass fractions in the droplet mode with high
MSEs. In the original IMPROVE formula, MSEs of these chemical species were
estimated using the multiple linear regression model according to the chemical
components in PM$_{2.5}$ and b$_{sp}$ from IMPROVE network, noting that significant mass
fractions of particle were in the condensation mode at the regional sites of IMPROVE
network and an urban site in U.S.A. (Cabada et al., 2004; Hand et al., 2002; Malm et
al., 2003). In contrast, in the present study most mass fractions of the dominant
chemical species (e.g. (NH$_4$)$_2$SO$_4$, NH$_4$NO$_3$ and OM) in the fine mode were
distributed in the droplet rather than condensation mode. These results suggested the
higher MSEs of (NH$_4$)$_2$SO$_4$, NH$_4$NO$_3$ and OM in the fine mode in this study were



likely due to their significant mass fractions in the droplet mode. In fact, the MSE of
fine soil in the IMPROVE formulas would represent the MSE of the bulk mode rather
than the fine mode (Hand and Malm, 2007). The average MSEs of the bulk mode was
$1.0\pm0.2$ $m^2$ $g^{-1}$ in this study, which was similar to that in the IMPROVE formulas.

On annual average, the estimated MSEs of $(NH_4)_2SO_4$, $NH_4NO_3$ and OM were

$4.7\pm0.6$, $4.8\pm0.9$ and $5.3\pm0.2$ $m^2$ $g^{-1}$ in the droplet mode, and were $2.1\pm0.5$, $2.3\pm0.8$
and $2.7\pm0.1$ $m^2$ $g^{-1}$ in the condensation mode, respectively, which were lower than
those in the large mode (similar to droplet mode) and were slightly lower than those
in the small mode (similar to condensation mode) in the revised IMPROVE formula
(Pitchford et al., 2007). Theoretically, the highest MSEs of $(NH_4)_2SO_4$, $NH_4NO_3$ and
OM would be found in about 0.55 μm in mass median geometric diameters (MMGD)
at the wavelength 550 nm according to Mie theory. However, the MMADs of
$(NH_4)_2SO_4$, $NH_4NO_3$ and OM were 0.76 - 0.80 μm (equal to about 0.60-0.64 μm in
MMGD) in the droplet mode and were 0.21 μm (equal to about 0.16-0.18 μm in
MMGD) in the condensation mode in this study, which were larger than 0.50 μm in
MMGD in the large mode and were lower than 0.20 μm in MMGD in the small mode
in the revised IMPROVE formula. Thus, the higher MMGDs in the droplet mode and
the lower MMGDs of $(NH_4)_2SO_4$, $NH_4NO_3$ and OM in the condensation mode in this
study maybe result in their lower MSEs compared with those in the revised
IMPROVE formula. In addition, the underestimated $b_{sp}$ would also result in
underestimating their MSEs in the condensation and droplet modes in this study.

Although the contribution of EC to $b_{sp}$ was not considered in the IMPROVE

formulas, its mass extinction efficiency (10 $m^2$ $g^{-1}$) considered both scattering and
absorption abilities (Hand and Malm, 2007). In fact, the theoretical average mass
absorption efficiency (MAE) of EC in fine particle was 7.5 $m^2$ $g^{-1}$ at the wavelength



550 nm (Wu et al., 2016). Thus, mass extinction efficiency of EC was also about 10
$m^2$ $g^{-1}$ in this study, suggesting the estimated EC MSEs were comparable with the
IMPROVE formulas. The estimated MSEs of coarse particle was 0.5±0.0 $m^2$ $g^{-1}$,
which was also comparable with the value of 0.6 $m^2$ $g^{-1}$ in the IMPROVE formulas.
Noticeably, sea salt was mainly distributed in the coarse mode rather than droplet
mode in this study. In addition, the unidentified fraction with large mass fraction and
the high MSE in the fine mode was not considered in the IMPROVE formulas,
although it accounted for a significant contribution of the estimated $b_{sp}$ in this study
(Fig. 7). In conclusion, EC and unidentified fraction rather than sea salt should be
considered in estimating $b_{sp}$, especially when EC and unidentified fraction accounted
for significant mass fractions of fine particles.
**3.3.2 Impact of size distribution on particle and chemical species MSE**
As discussed in section 3.3.1, seasonal average MSEs in the coarse mode
fluctuated in a small range of 0.4-0.5 $m^2$ $g^{-1}$, while those in the fine mode in a slightly
larger range of 3.5-3.9 $m^2$ $g^{-1}$, but the percentage changes are in similar magnitudes
(10-20%). Only variations of fine particle MSE were discussed below as an example.
It is worth to mention that fine particle MSE increased with its mass concentrations in
IMPROVE network (Lowenthal and Kumar, 2004), but such a phenomenon was not
founded in the present study. As shown in Fig. 8, the seasonal variations of fine
particle MSE were mainly caused by particle fractions in the size range of 0.4-0.9 μm,
which belong to the droplet mode. In this mode, the MSEs of $(NH_4)_2SO_4$ and $NH_4NO_3$
and OM were higher while those of the other chemical species were lower than the
overall particle MSE. Note that the overall particle MSE depends on the mass
concentrations and MSEs of individual chemical components. Thus, the seasonal





average MSEs of fine particle were dominated by the seasonal average mass fractions
and associated MSEs of $(NH_4)_2SO_4$ and $NH_4NO_3$ and OM in the droplet mode.

*Insert Figure 8*


The sum of the products of seasonal average mass concentration and MSEs of

the above three chemical species in the droplet mode was 1.8, 2.1, 2.3 and 2.5 $m^2$ $g^{-1}$
in spring, summer, autumn and winter, respectively. As expected, the seasonal
variations of fine particle MSE followed the sequences of winter (3.9±0.2 $m^2$ $g^{-1}$) >
autumn (3.8±0.2 $m^2$ $g^{-1}$) > summer (3.6±0.2 $m^2$ $g^{-1}$) > spring (3.5±0.1 $m^2$ $g^{-1}$).
Noticeably, fine particle MSE was determined by the average MSEs of the dominant
chemical species, rather than their mass fractions which were much smaller than 1.0.

Different from the approach used for fine particle MSE, the MSEs of $(NH_4)_2SO_4$,

$NH_4NO_3$ and OM in the droplet mode were determined using measurement-based
their mass size distributions prescribed as log-normal size distributions. The three
parameters describing the log-normal size distributions include mass concentration (in
the range of 0.43 - 2.1 μm), MMAD and standard deviation (σ). Thus, the MSEs of
$(NH_4)_2SO_4$, $NH_4NO_3$ and OM should depend on their MMADs and σ values. Seasonal
average σ values of $(NH_4)_2SO_4$, $NH_4NO_3$ and OM were in the range of 0.18-0.21,
0.18-0.21 and 0.22-0.26, respectively, while the corresponding MMADs in the range
of 0.72-0.92, 0.75-0.90 and 0.73-0.78 μm, respectively (Fig. 9). Generally, the
seasonal average MSEs of $(NH_4)_2SO_4$, $NH_4NO_3$ and OM in the droplet mode were



higher with the lower $\sigma$ values (or MMADs) when MMADs ($\sigma$ values) were close.
However, the MSE of OM in summer was 5.2 $m^2\ g^{-1}$, which was lower than 5.3 $m^2\ g^{-1}$
in autumn, although $\sigma$ values and MMADs in summer were lower than those in
autumn. This was mainly related with the evident fluctuation the MSE of OM in the
range of 0.6-0.7 $\mu m$.

*Insert Figure 9*


In conclusion, the fine particle MSE was determined by the sum of the products

of average mass fractions and MSEs of $(NH_4)_2SO_4$ and $NH_4NO_3$ and OM in the
droplet mode. The MSEs of the above three chemical species in the droplet mode
depended on both their $\sigma$ value and MMADs. Generally, fine particle MSE mainly
related with OM due to its high mass and MSE in the droplet mode in urban
Guangzhou.
**4. Summary and implication**

Size- and chemically-resolved particle number and mass concentration were

measured in urban Guangzhou in different seasons during 2015-2016 and the data
were used to estimate particle MSE. $SO_4^{2-}$ and $NH_4^+$ mainly distributed in the droplet
mode, EC in both condensation and droplet modes, and bulk particle, $NO_3^-$, OC, $Na^+$,
$Ca^{2+}$ and $Cl^-$ in both droplet and coarse modes. The estimated $b_{sp}$ can represent 85-89%
of the measured $b_{sp}$ based on the size-segregated chemical compositions according to
ISORROPIA II thermodynamic equilibrium model and Mie theory model. The largest

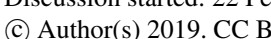



contributors to $b_{sp}$ were the chemical species in the droplet mode with the highest
MSEs.

MSEs of the dominant chemical species were noticeably different in this study

than those in the original and revised IMPROVE formulas. The MSEs of $(NH_4)_2SO_4$,
$NH_4NO_3$ and OM in the fine mode were higher than those in the original IMPROVE
formula, and in the droplet mode were lower than those in the revised IMPROVE
formula. In any case, $b_{sp}$ would be underestimated in urban Guangzhou using the
original or revised IMPROVE formulas because the unidentified chemical species
(and associated mass fractions) in the droplet mode accounted for a large fraction of
$b_{sp}$ and this portion was not included in these formulas. Moreover, MSEs of chemical
species would be overestimated in the original and revised IMPROVE formulas using
multiple linear regression model when the unidentified species was ignored. In
addition, sea salt was found in the coarse mode in this study, differing from the set up
in the IMPROVE formulas which is in the droplet mode. It can be concluded that the
estimated $b_{sp}$ in Guangzhou based on the revised IMPROVE formula would have
large biases, even though good correlations between estimated and measured $b_{sp}$ was
found.

MSEs of fine particles are controlled by the relative mass fractions of the

dominant chemical components (e.g., $(NH_4)_2SO_4$, $NH_4NO_3$ and OM) and associated
size distributions (e.g. σ and MMAD). Localized $b_{sp}$ formulas are thus needed for
better estimating particle MSE because particle size distributions of individual
chemical species vary significantly in space and time.



*Data availability. Data used in this study are available from Jun Tao (taojun@scies.org).*
*Competing interests. The authors declare that they have no conflict of interest.*

## Acknowledgments

This study was supported by the National Natural Science Foundation of China
(No.41475119 and 41603119). Original data are available from the corresponding
authors.

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





**Table 1 Summary of sampling information in urban Guangzhou**

| Season | Date | Sample type | Sampler | Sample duration | Sample number |
|---|---|---|---|---|---|
| Summer | 15 July- 6 August in 2015 | Size-segregated samples | Anderson 8-stage air samplers | 24h | 23 sets |
| Autumn | 15 October- 5 November in 2015 | | | 24h | 22 sets |
| Winter | 4-20 January - 19-22 February in 2016 | | | 24h | 21 sets |
| Spring | 8-20 April and 4-14 May in 2016 | | | 48h | 10 sets |
| Summer | 15 July- 6 August in 2015 | $PM_{2.5}$ and $PM_{10}$ samples | GSAPM | 24 h | 23 sets |
| Autumn | 15 October- 5 November in 2015 | | | | 22 sets |
| Winter | 4-20 January - 19-22 February in 2016 | | | | 21 sets |
| Spring | 8-20 April and 4-14 May in 2016 | | | | 20 sets |





## Table 2 Summary of chemical compositions concentrations in the different modes in urban Guangzhou

| Chemical composition | Size mode | Annual MMAD (μm) | Annual Mass (μg m⁻³) | Spring MMAD (μm) | Spring Mass (μg m⁻³) | Summer MMAD (μm) | Summer Mass (μg m⁻³) | Autumn MMAD (μm) | Autumn Mass (μg m⁻³) | Winter MMAD (μm) | Winter Mass (μg m⁻³) |
|---|---|---|---|---|---|---|---|---|---|---|---|
| PM | Condensation | 0.21±0.00 | 25.5±10.1(42±8%) | 0.21±0.00 | 25.3±7.0(40±7%) | 0.21±0.00 | 23.1±4.9(50±7%) | 0.21±0.00 | 30.8±11.8(44±6%) | 0.21±0.00 | 22.5±11.7(38±6%) |
|  | Droplet | 0.78±0.07 | 29.1±11.8(48±7%) | 0.87±0.13 | 31.9±8.7(50±8%) | 0.78±0.05 | 20.4±8.0(42±8%) | 0.79±0.05 | 35.6±9.7(46±4%) | 0.79±0.05 | 30.6±13.2(52±5%) |
|  | Coarse | 4.57±0.42 | 5.7±2.3(10±2%) | 4.37±0.37 | 6.6±3.0(10±3%) | 4.47±0.35 | 4.0±1.3(8±1%) | 4.90±0.46 | 7.0±1.9(10±2%) | 4.47±0.24 | 5.7±2.2(10±2%) |
| SO₄²⁻ | Condensation | 0.21±0.00 | 1.0±0.5(12%) | 0.21±0.00 | 0.9±0.3(10%) | 0.21±0.00 | 0.9±0.3(15%) | 0.21±0.00 | 1.4±0.5(13%) | 0.21±0.00 | 1.4±0.5(13%) |
|  | Droplet | 0.80±0.08 | 6.5±2.9(77%) | 0.86±0.07 | 7.3±2.3(79%) | 0.79±0.07 | 4.9±2.6(75%) | 0.77±0.08 | 8.5±2.6(75%) | 0.82±0.08 | 5.8±2.7(79%) |
|  | Coarse | 4.17±0.44 | 0.9±0.6(11%) | 4.34±0.59 | 0.9±0.6(11%) | 4.09±0.16 | 0.6±0.3(10%) | 4.08±0.22 | 1.4±0.8(12%) | 4.20±0.59 | 0.8±0.5(11%) |
| NO₃⁻ | Condensation | 0.21±0.00 | 0.4±0.3(10%) | 0.21±0.00 | 0.4±0.2(6%) | 0.21±0.00 | 0.2±0.2(9%) | 0.21±0.00 | 0.4±0.3(10%) | 0.21±0.00 | 0.6±0.3(13%) |
|  | Droplet | 0.85±0.21 | 2.2±2.2(46%) | 0.87±0.07 | 3.2±2.1(51%) | 0.94±0.35 | 0.8±0.5(35%) | 0.80±0.09 | 2.1±1.7(39%) | 0.80±0.07 | 3.2±2.9(63%) |
|  | Coarse | 4.38±0.61 | 1.8±1.4(44%) | 4.47±0.62 | 2.4±1.2(43%) | 4.15±0.52 | 1.3±0.7(56%) | 4.36±0.31 | 2.4±1.7(51%) | 4.74±0.76 | 1.3±1.5(24%) |
| NH₄⁺ | Condensation | 0.21±0.00 | 0.2±0.2(6%) | 0.21±0.00 | 0.2±0.1(6%) | 0.21±0.00 | 0.1±0.1(5%) | 0.21±0.00 | 0.2±0.2(7%) | 0.21±0.00 | 0.2±0.2(6%) |
|  | Droplet | 0.76±0.13 | 2.4±1.5(89%) | 0.86±0.17 | 2.8±1.1(89%) | 0.70±0.11 | 1.4±1.1(91%) | 0.73±0.12 | 3.1±1.4(90%) | 0.82±0.10 | 2.5±1.7(86%) |
|  | Coarse | 3.25±0.69 | 0.1±0.1(5%) | 3.13±1.16 | 0.2±0.2(6%) | 3.36±0.68 | 0.0±0.0(4%) | 3.01±0.23 | 0.1±0.1(3%) | 3.45±0.70 | 0.2±0.1(8%) |
| OC | Condensation | 0.21 | 1.2±0.6(13±4%) | 0.21±0.00 | 1.4±0.4(19±4%) | 0.21±0.00 | 0.8±0.3(11±4%) | 0.21±0.00 | 1.6±0.5(14±2%) | 0.21±0.00 | 1.2±0.6(13±4%) |
|  | Droplet | 0.76±0.07 | 5.5±2.4(62±9%) | 0.73±0.06 | 3.9±1.6(51±6%) | 0.77±0.07 | 4.1±1.3(63±9%) | 0.78±0.06 | 6.9±2.0(58±5%) | 0.75±0.08 | 6.5±2.6(69±7%) |
|  | Coarse | 3.73±0.58 | 2.2±1.1(25±8%) | 3.99±0.25 | 2.2±0.7(30±3%) | 3.50±0.73 | 1.7±0.9(26±9%) | 4.14±0.24 | 3.3±1.3(28±4%) | 3.44±0.39 | 1.7±0.9(18±8%) |
| EC | Condensation | 0.21±0.00 | 0.1±0.1(11%) | 0.21±0.00 | 0.1±0.0(9%) | 0.21±0.00 | 0.0±0.0(5%) | 0.21±0.00 | 0.1±0.1(16%) | 0.21±0.00 | 0.1±0.0(11%) |
|  | Droplet | 0.66±0.08 | 2.0±1.0(55±7%) | 0.65±0.08 | 1.8±0.8(54±9%) | 0.61±0.08 | 1.3±0.5(50±5%) | 0.71±0.04 | 2.7±0.9(62±6%) | 0.67±0.07 | 2.1±0.9(54±5%) |
|  | Coarse | 3.69±0.65 | 0.5±0.3(14±7%) | 3.54±0.61 | 0.3±0.2(10±6%) | 3.48±0.52 | 0.5±0.3(18±6%) | 4.17±0.24 | 0.6±0.2(14±5%) | 3.50±0.75 | 0.4±0.3(11±8%) |
| Na⁺ | Condensation | 0.21±0.00 | 0.1±0.1(11%) | 0.21±0.00 | 0.1±0.0(9%) | 0.21±0.00 | 0.0±0.0(5%) | 0.21±0.00 | 0.1±0.1(16%) | 0.21±0.00 | 0.1±0.0(11%) |
|  | Droplet | 0.86±0.12 | 0.4±0.2(48%) | 0.84±0.10 | 0.3±0.2(48%) | 0.96±0.11 | 0.4±0.1(45%) | 0.81±0.09 | 0.4±0.3(52%) | 0.80±0.11 | 0.3±0.2(48%) |
|  | Coarse | 3.75±0.38 | 0.4±0.3(41%) | 3.90±0.63 | 0.3±0.2(43%) | 3.60±0.19 | 0.6±0.4(50%) | 3.64±0.27 | 0.3±0.3(32%) | 3.94±0.38 | 0.3±0.2(41%) |
| K⁺ | Condensation | 0.21±0.00 | 0.1±0.0(13%) | 0.21±0.00 | 0.0±0.0(10%) | 0.21±0.00 | 0.1±0.0(16%) | 0.21±0.00 | 0.1±0.0(12%) | 0.21±0.00 | 0.1±0.0(12%) |
|  | Droplet | 0.69±0.08 | 0.3±0.2(78%) | 0.76±0.07 | 0.3±0.1(76%) | 0.64±0.08 | 0.3±0.1(72%) | 0.67±0.07 | 0.4±0.2(87%) | 0.73±0.06 | 0.4±0.2(77%) |
|  | Coarse | 3.74±0.51 | 0.0±0.0(9%) | 3.94±0.40 | 0.1±0.0(14%) | 3.74±0.64 | 0.0±0.0(12%) | 3.30±0.38 | 0.0±0.0(1%) | 3.78±0.35 | 0.0±0.0(11%) |
| Ca²⁺ | Condensation | 0.21±0.00 | 0.2±0.0(4%) | 0.21±0.00 | 0.1±0.0(7%) | 0.21±0.00 | 0.0±0.0(4%) | 0.21±0.00 | 0.0±0.0(3%) | 0.21±0.00 | 0.0±0.0(5%) |
|  | Droplet | 0.91±0.12 | 0.2±0.1(24%) | 0.88±0.13 | 0.3±0.1(36%) | 1.00±0.11 | 0.3±0.1(30%) | 0.81±0.10 | 0.2±0.1(16%) | 0.92±0.09 | 0.2±0.1(21%) |
|  | Coarse | 4.57±0.54 | 0.8±0.4(72%) | 5.02±0.58 | 0.6±0.2(57%) | 4.10±0.42 | 0.7±0.3(66%) | 4.72±0.47 | 1.1±0.5(81%) | 4.73±0.38 | 0.7±0.3(74%) |
| Cl⁻ | Condensation | 0.21±0.00 | 0.2±0.0(5%) | 0.21±0.00 | 0.2±0.0(5%) | 0.21±0.00 | 0.0±0.0(2%) | 0.21±0.00 | 0.0±0.0(5%) | 0.21±0.00 | 0.0±0.0(10%) |
|  | Droplet | 0.89±0.13 | 0.2±0.3(24%) | 0.89±0.10 | 0.7±0.7(37%) | 0.92±0.20 | 0.0±0.0(9%) | 0.89±0.05 | 0.0±0.0(17%) | 0.85±0.08 | 0.2±0.2(42%) |
|  | Coarse | 3.77±0.35 | 0.4±0.4(71%) | 3.97±0.12 | 0.8±0.4(58%) | 3.70±0.23 | 0.4±0.3(89%) | 3.72±0.21 | 0.3±0.2(78%) | 3.80±0.50 | 0.4±0.6(48%) |



Atmospheric Chemistry and Physics Discussions — Open Access

**Table 3 Summary of the estimated MSEs of particle and the dominant chemical species in urban Guangzhou**

| Chemical species | Size mode | Annual MMAD (µm) | Annual MSE (m²g⁻¹) | Spring MMAD (µm) | Spring MSE (m²g⁻¹) | Summer MMAD (µm) | Summer MSE (m²g⁻¹) | Autumn MMAD (µm) | Autumn MSE (m²g⁻¹) | Winter MMAD (µm) | Winter MSE (m²g⁻¹) |
|---|---|---|---|---|---|---|---|---|---|---|---|
| PM* | Condensation | 0.21±0.00 | 2.1±0.2 | 0.21±0.00 | 1.9±0.2 | 0.21±0.00 | 2.0±0.1 | 0.21±0.00 | 2.1±0.1 | 0.21±0.00 | 2.2±0.2 |
| | Droplet | 0.78±0.07 | 4.3±0.2 | 0.87±0.13 | 4.0±0.1 | 0.78±0.05 | 4.2±0.1 | 0.74±0.06 | 4.3±0.2 | 0.79±0.05 | 4.4±0.2 |
| | Coarse | 4.57±0.42 | 0.5±0.0 | 4.37±0.37 | 0.6±0.1 | 4.47±0.35 | 0.5±0.0 | 4.90±0.46 | 0.5±0.0 | 4.47±0.24 | 0.5±0.0 |
| | Fine mode** | | 3.7±0.2 | | 3.5±0.1 | | 3.6±0.2 | | 3.8±0.2 | | 3.9±0.2 |
| $(NH_4)_2SO_4$ | Condensation | 0.21±0.00 | 2.1±0.5 | 0.21±0.00 | 1.9±0.6 | 0.21±0.00 | 2.6±0.2 | 0.21±0.00 | 2.0±0.5 | 0.21±0.00 | 1.9±0.5 |
| | Droplet | 0.79±0.17 | 4.7±0.6 | 0.92±0.13 | 4.3±0.3 | 0.74±0.20 | 4.8±0.6 | 0.72±0.16 | 4.9±0.7 | 0.84±0.13 | 4.6±0.7 |
| | Fine mode | | 4.4±0.8 | | 4.1±0.4 | | 4.5±0.6 | | 4.6±0.8 | | 4.3±0.9 |
| $NH_4NO_3$ | Condensation | 0.21±0.00 | 2.3±0.8 | 0.21±0.00 | 2.0±0.8 | 0.21±0.00 | 2.9±0.3 | 0.21±0.00 | 2.6±1.0 | 0.21±0.00 | 2.3±0.7 |
| | Droplet | 0.80±0.16 | 4.8±0.9 | 0.90±0.18 | 4.5±0.8 | 0.77±0.17 | 4.9±0.8 | 0.75±0.13 | 5.1±1.0 | 0.82±0.14 | 4.7±0.8 |
| | Fine mode | | 4.5±1.5 | | 4.2±1.2 | | 4.7±0.9 | | 4.9±2.0 | | 4.4±1.3 |
| OM | Condensation | 0.21±0.00 | 2.7±0.1 | 0.21±0.00 | 2.5±0.1 | 0.21±0.00 | 2.8±0.2 | 0.21±0.00 | 2.6±0.1 | 0.21±0.00 | 2.8±0.1 |
| | Droplet | 0.76±0.07 | 5.3±0.2 | 0.73±0.06 | 5.4±0.1 | 0.77±0.07 | 5.2±0.2 | 0.78±0.06 | 5.3±0.2 | 0.75±0.08 | 5.5±0.2 |
| | Coarse | 3.73±0.58 | 0.8±0.1 | 3.99±0.25 | 0.8±0.0 | 3.50±0.73 | 0.8±0.1 | 4.14±0.24 | 0.7±0.0 | 3.44±0.39 | 0.8±0.1 |
| | Fine mode | | 4.6±0.3 | | 4.4±0.2 | | 4.6±0.2 | | 4.5±0.1 | | 4.9±0.3 |
| EC | Condensation | 0.21±0.00 | 2.9±0.1 | 0.21±0.00 | 2.9±0.1 | 0.21±0.00 | 2.9±0.1 | 0.21±0.00 | 3.0±0.1 | 0.21±0.00 | 2.9±0.1 |
| | Droplet | 0.66±0.08 | 2.3±0.2 | 0.65±0.08 | 2.3±0.2 | 0.61±0.08 | 2.3±0.2 | 0.71±0.04 | 2.2±0.1 | 0.67±0.07 | 2.3±0.2 |
| | Coarse | 3.69±0.65 | 0.4±0.0 | 3.54±0.61 | 0.4±0.0 | 3.48±0.52 | 0.4±0.0 | 4.17±0.24 | 0.4±0.0 | 3.50±0.75 | 0.5±0.0 |
| | Fine mode | | 2.6±0.1 | | 2.6±0.1 | | 2.6±0.2 | | 2.5±0.1 | | 2.6±0.1 |
| Crustal element oxides | Condensation | 0.21±0.00 | 0.7±0.0 | 0.21±0.00 | 0.7±0.0 | 0.21±0.00 | 0.7±0.1 | 0.21±0.00 | 0.7±0.0 | 0.21±0.00 | 0.7±0.0 |
| | Droplet | 0.91±0.12 | 2.9±0.2 | 0.88±0.13 | 3.0±0.2 | 1.00±0.11 | 2.9±0.2 | 0.81±0.10 | 2.8±0.2 | 0.92±0.09 | 2.8±0.2 |
| | Coarse | 4.57±0.54 | 0.4±0.0 | 5.02±0.58 | 0.4±0.0 | 4.10±0.42 | 0.5±0.0 | 4.72±0.47 | 0.4±0.0 | 4.73±0.38 | 0.4±0.0 |
| Unidentified | Condensation | 0.21±0.00 | 1.3±0.2 | 0.21±0.00 | 1.2±0.4 | 0.21±0.00 | 1.2±0.2 | 0.21±0.00 | 1.4±0.1 | 0.21±0.00 | 1.3±0.2 |
| | Droplet | 0.85±0.26 | 3.8±0.6 | 1.00±0.20 | 3.5±0.8 | 0.74±0.44 | 3.9±0.9 | 0.84±0.10 | 3.9±0.2 | 0.90±0.20 | 3.7±0.4 |
| | Coarse | 5.74±1.52 | 0.4±0.1 | 4.55±0.71 | 0.5±0.1 | 6.46±1.14 | 0.4±0.1 | 6.33±1.62 | 0.4±0.1 | 4.91±0.90 | 0.5±0.1 |
| | Fine mode | | 3.1±0.8 | | 2.9±0.9 | | 2.6±1.0 | | 3.3±0.3 | | 3.1±0.5 |
| NaCl | Coarse | 4.88±0.41 | 0.5±0.1 | 5.14±0.70 | 0.5±0.1 | 4.49±0.38 | 0.6±0.0 | 5.38±0.43 | 0.5±0.0 | 4.66±0.65 | 0.5±0.0 |

*PM: Particulate matter; **Fine mode = sum of condensation and droplet modes





**List of Figures**





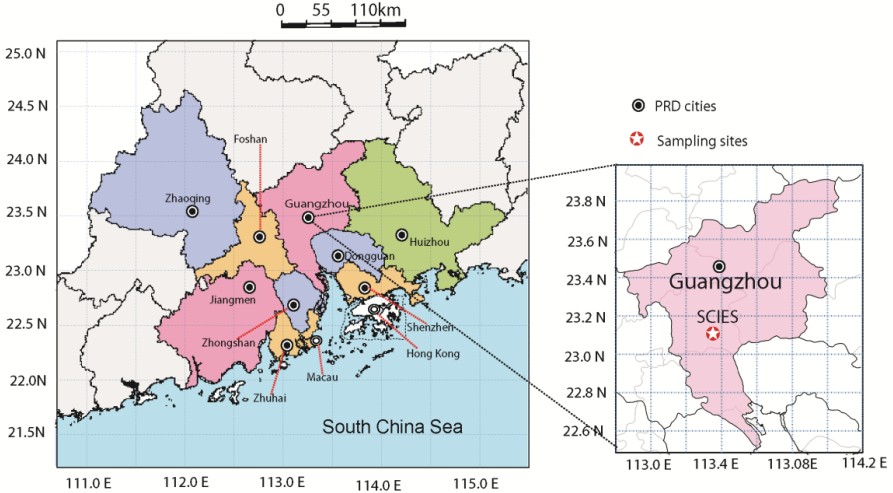

Fig. 1. The sampling locations in urban Guangzhou in the PRD region of China.





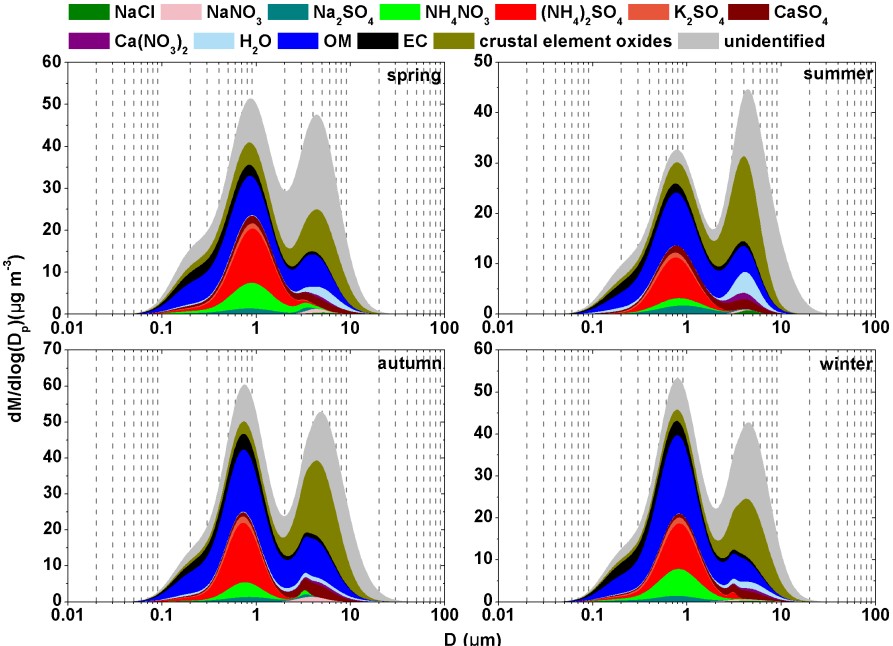

Fig. 2. Continuous log-normal size distributions of chemical species mass
concentrations in four seasons (dlogD$_a$=0.01μm).





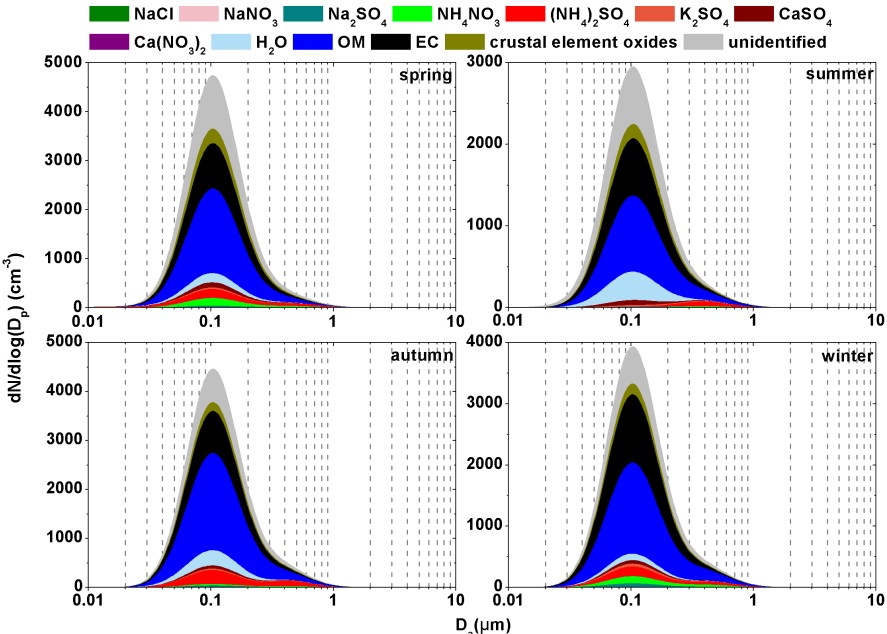

Fig. 3. Continuous log-normal size distributions of the estimated chemical species

number concentrations in four seasons (dlogD$_a$=0.01μm).





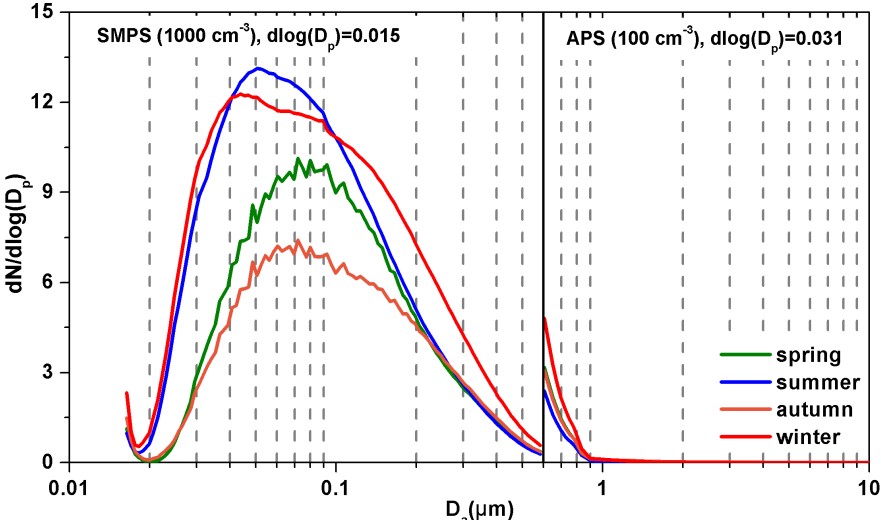

Fig. 4. Continuous log-normal size distributions of the measured particle number

concentrations in four seasons.




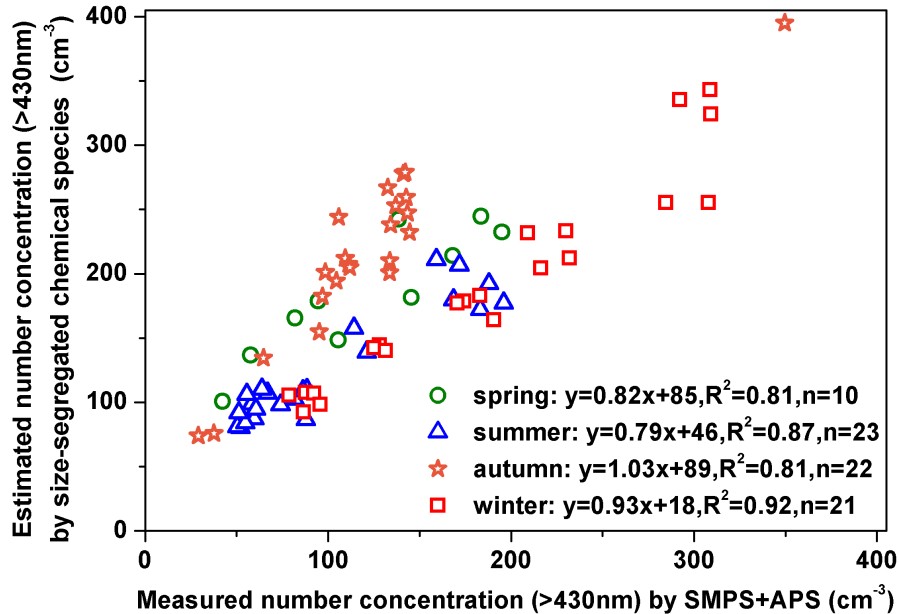

Fig. 5. Correlations between the estimated and SMPS- and APS-measured particle

number concentrations in four seasons.





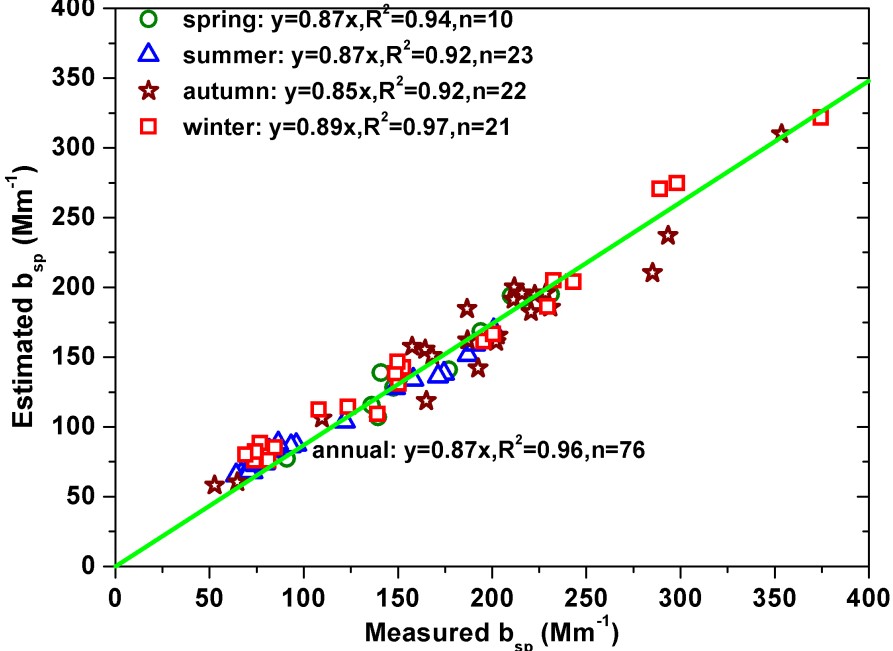

Fig. 6. Correlations between the measured and estimated $b_{sp}$ in four seasons.



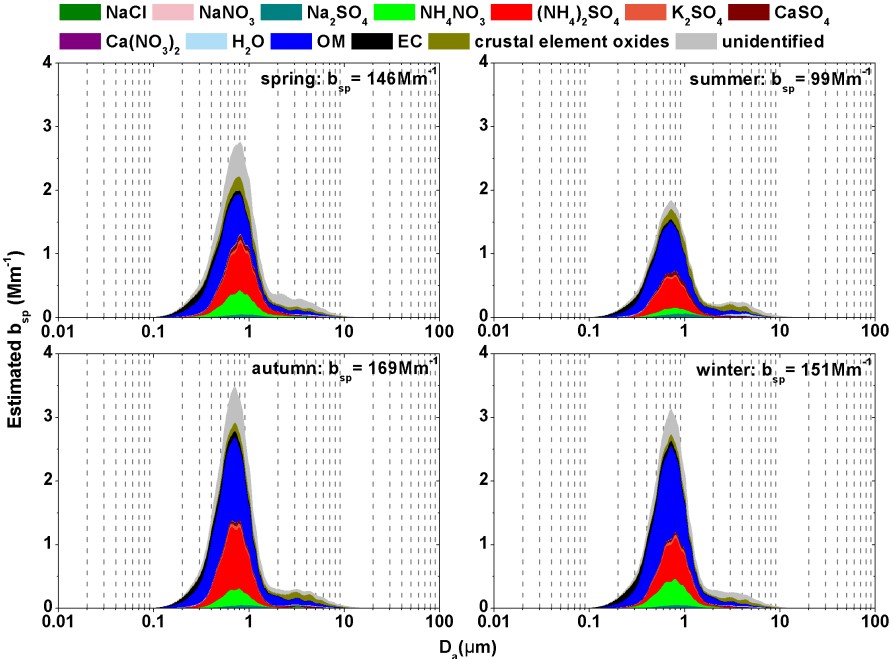

Fig. 7. The contributions of continuous log-normal size distributions of chemical species on the estimated $b_{sp}$ in four seasons ($dlogD_a$=0.01μm).



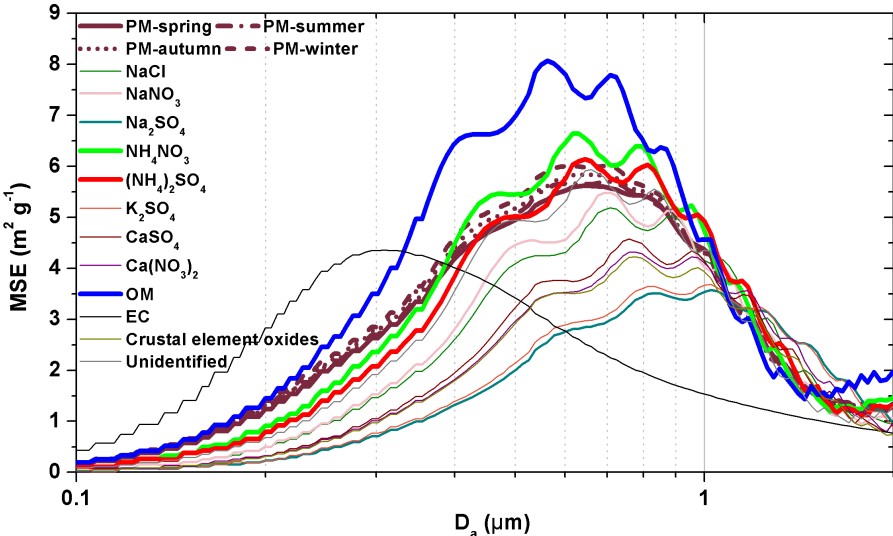

Fig. 8. Continuous log-normal size distributions of fine particle MSEs in four seasons and the MSEs of chemical species.





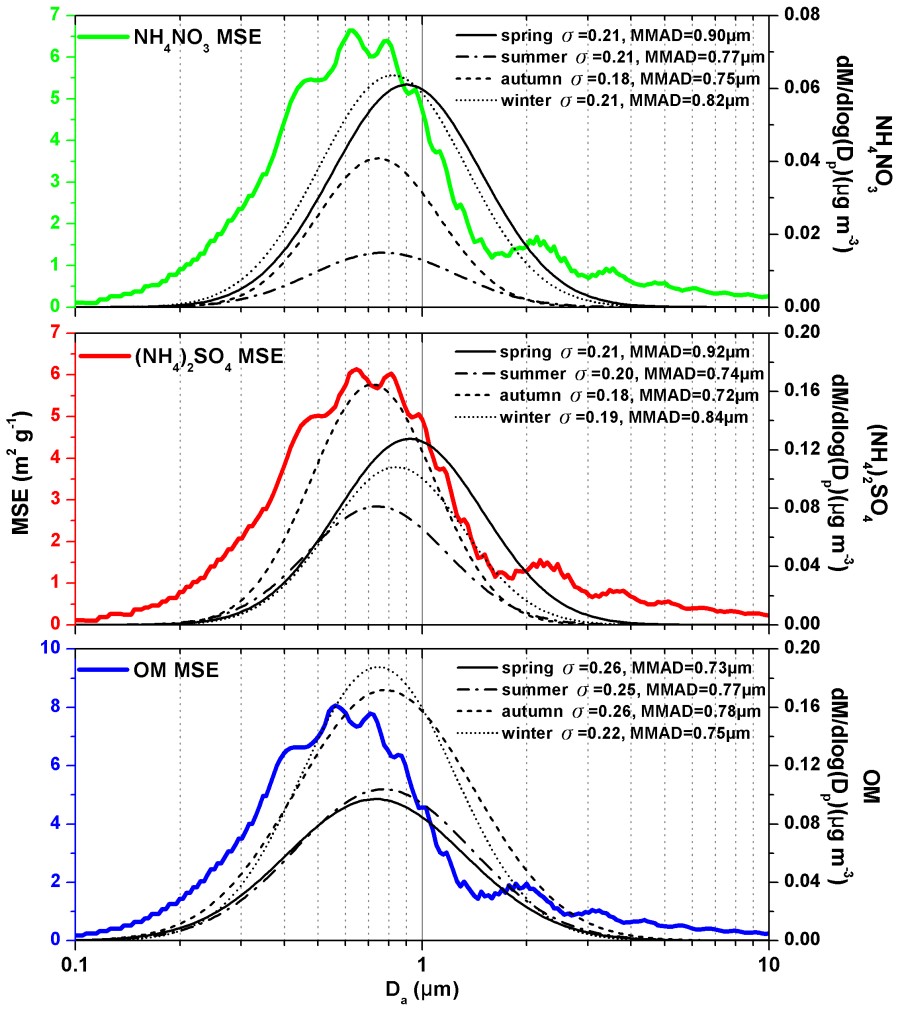

Fig. 9. Continuous log-normal size distributions of $(NH_4)_2SO_4$ (a), $NH_4NO_3$ (b) and OM (c) mass concentrations and their σ values and MMADs in the droplet mode.