# Peer review of "Impact of particle number and mass size distributions of major chemical components on particle mass scattering efficiency in urban Guangzhou of South China"

_Atmospheric Chemistry and Physics, 2018_

## Referee Comment (RC1) · Anonymous Referee #1 · 15 Mar 2019

The authors investigate key factors affecting mass scattering efficiencies by using particle size distribution data, PM2.5 and PM10 bulk data, and measured light scattering coefficients. Measurements were made at an urban site in China. The authors characterized the size, composition, and mass scattering efficiencies of the major aerosol modes and determined the major species contributions to mass and scattering for each mode. They used this information to calculate particulate mass scattering efficiencies for each mode. The study provides useful information on mass scattering efficiencies– parameters needed for visibility estimates and to relate mass to scattering for satellite

or modeling applications. The paper is fairly well organized and written but is in need of many clarifications, as described in the comments below.

Line 7: I'm concerned about reporting the condensation mode MMAD of 0.21 um. This just reflects the midpoint of the diameter bin. If this is the case, why not just report the midpoint of diameter bins for all the modes? Reporting it like data is meaningless.

Line 19: How is "fine" defined here?

Line 42: Define "IMPROVE" on first usage.

Line 45: MSE are important parameters not just for the IMPROVE equation, but for any application relating mass to optical properties.

Line 49: Include "based on an assumed size distribution" after "formula. . ."

Line 60: Yes, the second IMPROVE algorithm was developed for rural (very clean) areas, so it isn't a surprise that it doesn't perform well in urban areas.

Line 69: I think the reference here should be "Malm and Hand, 2007". Also, the efficiencies used in the second IMPROVE algorithm are based on an assumed size distribution and composition.

Line 73: Consider removing "According to Mie theory" because Mie theory doesn't specifically speak to the factors hindering the IMPROVE formulas. Line 75: Also, what about assumed hygroscopic growth curves in the IMPROVE algorithm?

Line 81: I think the authors mean "inline" when they say "online" data?

Line 88: Especially in urban areas.

Line 116: It would also help to include the other measurements in Table 1, such as the size distributions and nephelometer measurements.

Line 123: Do the authors mean "blank" instead of "background"?

Line 136: What is the expected size cut of the nephelometer? Are there expected

size-resolved losses from the tubing from the inlets that affect the size distribution and nephelometer measurements?

Line 140: Define "RH" at first usage (unless I missed it earlier).

Line 141: Were RH and temperature monitored? What were typical values?

Line 179: How were field blanks obtained?

Line 179: Why was OP so low from the 81mm filters?

Line 201-205: I am not sure of the rationale behind defining the condensation mode as the midpoint diameter of the smallest bin? If the MMAD of the mode is just assigned the midpoint diameter of the bin, what point is there in measuring any size distributions? The MMAD would just be the midpoint diameter of each bin which is meaningless. I don't think you can report the MMAD of the condensation mode if this is how you derive it.

Line 209: I assume this discussion is with respect to the technique by Dong et al. (2004)? It might be helpful to provide more detail here regarding this method, since many of the results depend on it. For example, how were collection efficiencies incorporated into this inversion?

Line 215: Is this size resolved mass from the thermodynamic model on the binned data or the fit data?

Line 217: A section on the DMA and APS size distribution analysis is needed. How was the APS calibrated? How was aerodynamic diameter converted to mobility diameter (or vice versa)? What is the response of the APS to particles of different density? How was density calculated?

Line 225: Is PM10 here the bulk gravimetric PM10 or the summed data from the impactor? Does this include water at the RH of the PM10 gravimetric measurement? Particle bound water can still exist for 40% RH.

[Figure]

Line 226-227: Units for MMAD?

Line 226: How were MMADs calculated for the 'continuous' lognormal data?

Line 226: Again, reporting an MMAD for the condensation mode is meaningless.

Line 229: Define PRD.

Line 237: Close to what?

Line 289: What about K+ in the fine mode?

Line 336: Coarse mode mass fractions also depend on other species. Do the authors mean their absolute concentrations rather than relative concentrations?

Line 339: Change title to "Closure of particle mass, number concentration, and bsp"

Line 345: Was sulfate fully neutralized for the duration of the study?

Line 355: Was 5% used here?

Line 362: I am not sure what is meant here by the "total"? How was "total" derived in this context?

Line 371: Same comment as the previous.

Line 382: Please provide more details regarding this method.

Line 384: This would be expected because of the diameter-cubed dependence between number and mass.

Line 386: What is the difference in the definition of the estimated NMAD of the number concentrations of individual species and the NAMD of particle number concentrations? (individual versus particle?). I think an issue here is that the constant 0.21 $\mu$m value is meaningless.

Line 392: What densities were used?

Line 397: The reasoning here isn't clear. The size segregated chemical mass species concentrations should be dry. Unless the authors mean that particle bound water was associated with a gravimetric measurement, the individual species mass do not include water.

Line 418-420: I am unclear as to why scattering efficiencies are being discussed here?

Line 421-423: This is the first discussion of these design flaws – are the authors referring to the single bin for the condensation mode?

Line 426: How much higher?

Line 429: How do the authors know that EC was internally mixed with OM or inorganic salts during this study?

Line 433-434: The reasoning here is unclear. What are the estimation errors and models?

Line 448: The authors need to provide more details on how they derived bsp. What refractive indices did they use, how did they calculate them, which number size distributions did they use, etc.

Line 455: Why "especially the inversion technique method"?

Line 460: What do the authors mean that OC was underestimated by the OC/EC protocol?

Line 468, 472: Do the authors mean "inline" data?

Line 502: What did the authors use for refractive indices for the "unidentified fraction"?

Line 517, 521: I am not sure what is meant by "particle and chemical species". What is the distinction?

Line 540: This points back to the previous comments as well. Was "particle MSE" estimated by summed bsp from individual species divided by summed particle mass,

or was bsp calculated for "particle", which then would require a "particle" refractive index? It would help if the authors provided details for how these things are calculated (see comment for line 448).

Line 577: Define MMGD

Line 626-627: Sentence is unclear.

Line 630: How were sigma values calculated?

Line 653: What does "bulk particle" mean?

Line 670: Sea salt in the IMPROVE formula is assumed to have a mass mean diameter of 2.5 um, so it is assumed to be in the coarse mode with the tail extending into the PM2.5 mode.

Figures and Tables: Table 2: Define size range of condensation, droplet and coarse modes. Again, reporting 0.21 um for all condensation mode MMAD is meaningless. Define "MMAD" in the caption.

Table 3: Define size range of condensation and droplet modes. Again, reporting 0.21 um for all condensation mode MMAD is meaningless. Define "MSE" and "MMAD" in the caption. Include wavelength and relative humidity (Dry = ?%) in the caption or subtitle.

Figure 1: Define "PRD" in the caption.

Figure 2: Was CaSO4 and Ca(NO3)2 subtracted out of the soil formula when using Ca to calculate soil? These figures suggest that EC mass size distributions are larger than OM distributions? Are these stacked? If so EC » SO4 but mass concentrations in Table 2 suggests this is not the case. The presentation is somewhat confusing. Keep the y-axis the same for all seasons for easier comparisons.

Figure 3: Similar comments as previous caption.

Figure 4: It would help to plot the APS data in terms of mass or volume instead of

number based on the size range- the larger modes would be more visible.

Figure 6: Include wavelength, relative humidity conditions, and size range. Is this total bsp?

Figure 7: Similar comments to figure 2. I don't understand how the mass of sulfate can be so much higher than EC yet the EC scattering is greater?

Figure 8: Define "MSE", "fine", wavelength, and relative humidity conditions in the caption.

Figure 9: Caption doesn't include any information on MSE. Also include wavelength, and relative humidity conditions in the caption.

---

## Referee Comment (RC2) · Anonymous Referee #2 · 22 Mar 2019

General

The paper presents calculations of mass scattering efficiencies (MSE) in Guangzhou, a polluted region in south China. The work is based on size-segregated measurements of aerosols with a multi-stage impactor, filter sampling, size distributions measured with an SMPS and an APS and scattering measured with a nephelometer. The main goal was to obtain new MSEs to be used in polluted air instead of those in the IMPROVE equation. That makes sense since the IMPROVE equation was derived from data collected in very different areas, mainly US national parks. The paper is traditional

aerosol science, there are no especially significant new findings but it is important to get these new MSEs published since they are a piece in improving air quality. The paper was easy to read, I did not find any major mistakes so I can recommend publishing it in ACP after making some small additions and corrections.

One thing that I recommend doing is to calculate your MSEs also by using multiple linear regression (MLR). Now you calculated them with a Mie model. That is fine and scientifically justified but it also has its uncertainties, for instance related to refractive indices etc. Your data is good for MLR and that would give another estimate for the MSEs. MLR is quick and easy to do – even with Excel – and that is also actually inversely the way air quality data would be used for estimating visibility from PM2.5 filter data. Doing that you would have an additional uncertainty estimate and a closure of MSEs.

Having done that I suggest you make an additional scatter plot and linear regressions of scattering coefficient calculated with the Mie-derived MSEs, with the MLR-derived MSEs and with IMPROVE MSEs vs. measured scattering coefficient. Now you have written in the text new MSEs and written how they differ from the IMPROVE MSEs but the full comparison for the Guanzhou air is missing, that would be the linear regressions I suggested. How well do the different MSEs predict the observed scattering?

Another thing I miss is equations. For example equations of how you calculated MSE, the mean diameters you are using and also chemistry: Did you dry the sampling air for the impactor? If not the particles are larger and get collected on the upper stages which affects the inverted size distributions and ultimately the Mie-modeled scattering. At least some discussion of this would be good.

Detailed comments L131 " ... geometric diameter (Dg) ..." The widely used meaning of Dg is the geometric mean diameter of a particle number size distribution. So use Dp. for the aerodynamic diameter use Da.

L137-138. Nephelometer: did yu calibrate it?

L197: expain the Mie model in a bit more detail

L201, define MMAD and give the formula L206 "limit of detection" is wrong here, that expression is related to concentration measurements

L248 "As expected" – why would you expect this?

L266 "NO3 mainly exists in the form of ammonium nitrate..." you have data on the inorganic ion concentrations but how did you calculate concentration of ammonium sulfate and ammonium nitrate? Give a couple of formulas.

L385 "NMAD" – give formula

L574 "mass median geometric diameter (MMGD)" I have never heard of. Define. Consider using some other descriptive diameter that has been presented in literature.

Fig. 4. Are the diameters of the ASMPS data and the APS data both aerodynamic or what? The gap is huge, try to explain it.

Fig 5. The numbers in the x and y axes cannot be true. In Guangzhou number concentrations are in the range of thousands, now the max concentration is about 400 /cc.

---

## Author Comment (AC1) · 10 May 2019

**Response to Reviewer #1**

We greatly appreciate the reviewer for providing the very detailed comments, which have helped us improve the paper quality significantly. We have addressed all of the comments carefully as detailed below. The original comments are in black and our replies are in blue.

Line 7: I'm concerned about reporting the condensation mode MMAD of 0.21 um. This just reflects the midpoint of the diameter bin. If this is the case, why not just report the midpoint of diameter bins for all the modes? Reporting it like data is meaningless.

We agree with this comment and thus have deleted the MMAD for the condensation mode.

Line 19: How is "fine" defined here?

"Fine particles" are defined here as those with aerodynamic diameter smaller than 2.1 µm because the cutoff size of the instrument is at 2.1 µm.

Line 42: Define "IMPROVE" on first usage.

We have revised the text as follows: "the original and revised empirical formulas from the Interagency Monitoring of Protected Visual Environments (IMPROVE) network".

Line 45: MSE are important parameters not just for the IMPROVE equation, but for any application relating mass to optical properties.

We have revised the text as follows: "MSEs of the chemical species are important parameters not only for building the relationships between chemical species and $b_{sp}$ (Hand and Malm, 2007), but also for relating particle mass to its optical properties (Lin et al., 2015; Titos et al., 2012)".

Line 49: Include "based on an assumed size distribution" after "formula…"

Text added as suggested.

Line 60: Yes, the second IMPROVE algorithm was developed for rural (very clean) areas, so it isn't a surprise that it doesn't perform well in urban areas.

We agree with this comment. However, the majority of the studies in China still used the revised formula in urban environment likely because the original IMPROVE formula evidently underestimated $b_{sp}$. It is thus needed to further assess the uncertainties in these formulas when applying to urban environments.

Line 69: I think the reference here should be "Malm and Hand, 2007". Also, the efficiencies used in the second IMPROVE algorithm are based on an assumed size distribution and composition.

Reference replaced as suggested.

Line 73: Consider removing "According to Mie theory" because Mie theory doesn't specifically speak to the factors hindering the IMPROVE formulas. Line 75: Also, what about assumed hygroscopic growth curves in the IMPROVE algorithm?

We have deleted the text "According to Mie theory". In this study, we only discussed closure of $b_{sp}$ under dry condition, not wet condition.

Line 81: I think the authors mean "inline" when they say "online" data?

We have replaced "online" with "inline" in all the places.

Line 88: Especially in urban areas.

We have revised the text as follows: "Knowledge gained from the present study will improve the assessments of air-quality and climate impact caused by atmospheric particles, especially in urban areas."

Line 116: It would also help to include the other measurements in Table 1, such as the size distributions and nephelometer measurements.

We have added the relevant instruments information in Table 1.

Line 123: Do the authors mean "blank" instead of "background"?

Revised as suggested.

Line 136: What is the expected size cut of the nephelometer? Are there expected size-resolved losses from the tubing from the inlets that affect the size distribution and nephelometer measurements?

We have added the following text in the revised paper to address this comment: "According to the method described in Kulkarni et al. (2011), particle losses in different sizes from the tube are plotted in Fig. S1. Generally, particle losses in the condensation (0.1-0.4 µm), droplet (0.4-2.1 µm) and coarse modes (2.1-10 µm) were less than 1.3%, 0.3% and 0.1%, respectively, suggesting that the particle losses from the tube were minimal. Ambient relative humidity (RH) and temperature were measured by an automatic meteorological station (Vaisala Company, Helsinki, Finland, model MAWS201) at the SCIES site, and the seasonal average of these two meteorological

parameters were 53-75 % and 15-29 °C, respectively."

[Figure]

Fig. S1: The estimated particle losses in different size from the tube.

Line 140: Define "RH" at first usage (unless I missed it earlier).

Defined as suggested.

Line 141: Were RH and temperature monitored? What were typical values?

We have added the following text in the revised paper: "Ambient relative humidity (RH) and temperature were measured by an automatic meteorological station (Vaisala Company, Helsinki, Finland, model MAWS201) at the SCIES site, and the seasonal average of these two meteorological parameters were 53-75 % and 15-29 °C, respectively."

Line 179: How were field blanks obtained?

We have added this description in the revised paper: "Moreover, 8 sets of blank samples were also collected for each of the size-segregated particle, $PM_{2.5}$ and $PM_{10}$ samples during the whole sampling period. Two sets of blank filters in each category were put in the samplers without flow for 24 h when seasonal field campaigns finished. The aerosol-loaded filter samples were stored in a freezer at -18 °C before analysis to prevent volatilization of particles."

Line 179: Why was OP so low from the 81mm filters?

Firstly, particle sample showed dot pattern (100-400 dots in every stage) in the size-segregated filters (81 mm filters). Secondly, carbon analyzer only analyzes one punch (0.526 $cm^2$), which contained 4-5 dots. Thirdly, total carbon loading in fine particle was not high (about 5-30 µg $m^{-3}$). If its concentration was distributed into 8 stages, then OC

and EC concentrations in each stage would be very low. In addition, OP concentration was much lower than OC and EC. Thus, the uncertainties in OC and EC concentrations would be larger using OP to separate OC and EC in each stage.

Line 201-205: I am not sure of the rationale behind defining the condensation mode as the midpoint diameter of the smallest bin? If the MMAD of the mode is just assigned the midpoint diameter of the bin, what point is there in measuring any size distributions? The MMAD would just be the midpoint diameter of each bin which is meaningless. I don't think you can report the MMAD of the condensation mode if this is how you derive it.

As mentioned above, we have deleted the statement of MMAD for the condensation mode throughout the whole manuscript.

Line 209: I assume this discussion is with respect to the technique by Dong et al. (2004)? It might be helpful to provide more detail here regarding this method, since many of the results depend on it. For example, how were collection efficiencies incorporated into this inversion?

The reviewer is right. Here we indeed refer to the technique of Dong et al. (2004). We have added the key formulas in in section 2.4.

We have also revised the text as follows: "Continuous size-distribution profiles of major chemical species are needed in order to accurately calculate $b_{sp}$ using Mie theory. To improve the resolution of $b_{sp}$, 401 bins were used for chemical species ranging from 10 nm to 100 μm, with a constant ratio between the adjacent size bins, defined as $\log_{10}(D_{a2}/D_{a1})=0.01$. Further increasing the number of size bins does not have any significant impact on the results, e.g., the changes in $b_{sp}$ are smaller than 1% even if the above ratio of 0.01 is replaced with 0.001. Continuous size-distribution profiles of major chemical species are obtained from the inversion of the measured mass concentration distribution in the size bins of the Anderson 8-stage air samplers, using the technique described in Dong et al. (2004). The key formulas to calculate the normal distribution of density function ($f(D, \mu, \sigma)$) were summarized as follows:

$$f(D, \mu, \sigma) = \frac{1}{\sqrt{2\pi}\sigma} e^{-\left(\frac{(D-\mu)^2}{2\sigma^2}\right)}$$

(6)

$$\mu = \bar{y} - \mu\bar{x} \tag{7}$$

$$\sigma = \frac{n\sum xy - \sum x \times \sum y}{n\sum x^2 - (\sum x)^2} \tag{8}$$

Where D is $\log(D_a)$, and μ and σ are the mean and standard deviation, respectively, of the $\log(D_a)$ in the different modes. x is the inverse function value of the cumulative probability of a standard normal distribution in each bin, y is logarithm of $D_a$ lower limit (e.g. 0.43, 0.65, 1.1, 2.1, 3.3, 4.7, 5.8 and 9.0 μm) in each bin. An example of the calculation process was demonstrated in supplementary.

However, this approach is not applicable for the condensation mode because there is only one size bin in this mode. To obtain the number concentrations of all the concerned chemical species in the condensation mode, MMADs ($=10^\mu$) of this mode are calculated according to:

$$MMADs = (D_{a1} \times D_{a2})^{0.5} \tag{9}$$

Where $D_{a1}$ and $D_{a2}$ represent the lower (0.10 μm, limits of detection of Anderson 8-stage air sampler) and upper (0.43 μm) boundaries of this size bin, respectively.

Line 215: Is this size resolved mass from the thermodynamic model on the binned data or the fit data?

We have revised the text as follows: "The ISORROPIA II model was run at the reserved mode (Fountoukis and Nenes, 2007) with input data of $K^+$, $Ca^{2+}$, $Mg^{2+}$, $NH_4^+$, $Na^+$, $SO_4^{2-}$, $NO_3^-$, $Cl^-$, RH (40%), and temperature (25°C), to estimate the size-resolved mass concentrations of $NaCl$, $NaNO_3$, $Na_2SO_4$, $NaHSO_4$, $NH_4Cl$, $NH_4NO_3$, $(NH_4)_2SO_4$, $NH_4HSO_4$, $K_2SO_4$, $KHSO_4$, $KNO_3$, $KCl$, $MgSO_4$, $Mg(NO_3)_2$, $MgCl_2$, $CaSO_4$, $Ca(NO_3)_2$, $CaCl_2$ and $H_2O$. Several of these chemical species had extremely low mass concentrations and were thus excluded from the calculation of $b_{sp}$. Generally, only $NaCl$, $NaNO_3$, $Na_2SO_4$, $NH_4NO_3$, $(NH_4)_2SO_4$, $K_2SO_4$, $Ca(NO_3)_2$, $CaSO_4$ and $H_2O$ were used to estimate $b_{sp}$ in this study."

Line 217: A section on the DMA and APS size distribution analysis is needed. How was the APS calibrated? How was aerodynamic diameter converted to mobility diameter (or vice versa)? What is the response of the APS to particles of different density? How was density calculated?

The measured particle number concentrations by SMPS and APS were used to assess the accuracy of the estimated particle number concentrations of chemical species in section 3.2.2. We have added the calibration procedure of APS in section 2.2 as follows: "APS was calibrated using 5 sizes solid spheres (polystyrene latex monodisperse)."

We have added the convert formula in section 2.4: "The measured particle number concentrations using SMPS in $D_p$ (similar to $D_g$) were converted to the particle number concentrations in aerodynamic diameter according to:

$$D_a = D_p / (\rho)^{0.5} \tag{6}$$
$$\rho = \frac{\sum_{chemical\ species} m_i}{\sum_{chemical\ species} \frac{m_i}{\rho_i}} \tag{7}$$

Where, $\rho$ represents the daily average densities of particle, $m_i$ is chemical species mass concentration in a bin, $\rho_i$ is chemical species density. The seasonal average densities of particle were calculated in Fig. S4."

[Figure]

Fig. S4: Continuous log-normal size distributions of seasonal average densities in four seasons.

Line 225: Is $PM_{10}$ here the bulk gravimetric $PM_{10}$ or the summed data from the impactor? Does this include water at the RH of the $PM_{10}$ gravimetric measurement? Particle bound water can still exist for 40% RH.

Here, $PM_{10}$ mass was the sum of the size-segregated mass concentrations. We revised the text as follows: "On annual average, $10\pm2\%$, $48\pm7\%$ and $42\pm8\%$ of total mass in the size-segregated samples were in the condensation, droplet and coarse modes, respectively, with the average MMADs being $0.78\pm0.07$ µm in the droplet mode and $4.57\pm0.42$ µm in the coarse mode." Yes, particles content a small amount of water even at RH=40% according to the ISORROPIA II model (as shown in Fig.2).

Line 226-227: Units for MMAD?

Units added for MMAD in the revised paper.

Line 226: How were MMADs calculated for the 'continuous' lognormal data?

We have added the key formulas for calculating MMADs in section 2.4. The formula in the droplet and coarse modes : MMADs $(=10^{\mu})$. The formula in the condensation mode: MMADs $=(D_{a1}\times D_{a2})^{0.5}$, where, $D_{a1}$ and $D_{a2}$ represent the lower (0.10 µm, limits of detection of Anderson 8-stage air sampler) and upper (0.43 µm) boundaries of this size bin, respectively. Here MMADs in the condensation mode were only used for estimating the continuous lognormal chemical species mass and number concentrations.

Line 226: Again, reporting an MMAD for the condensation mode is meaningless.

We have deleted the statements on the MMADs for the condensation mode in this manuscript. However, the MMADs in the condensation mode were still used to estimate the continuous lognormal chemical species mass and number concentrations.

Line 229: Define PRD.

We have added the text "the Pearl River Delta (PRD) region".

Line 237: Close to what?

We have revised the text as follows: "Seasonal average particle mass concentrations were evidently lower in summer than in the other seasons for the condensation and droplet modes, and were similar during spring, autumn and winter for all the three modes."

Line 289: What about $K^+$ in the fine mode?

We have added the size distribution of $K^+$ in Fig. S5.

[Figure]

Fig. S5: Continuous log-normal size distributions of $K^+$ in four seasons.

Line 336: Coarse mode mass fractions also depend on other species. Do the authors mean their absolute concentrations rather than relative concentrations?

Yes, coarse mode mass fraction also depended on chemical species especially OM and crustal element oxides. Here, $PM_{10}$ mass concentration was the absolute (not relative) concentration. We have revised the text as follows: "Annual average $PM_{10}$ concentrations (46 µg m$^{-3}$) in 2015-2016 in the PRD region were about 40% lower than that (76 µg m$^{-3}$) in 2006-2007, which further supported the above hypothesis."

Line 339: Change title to "Closure of particle mass, number concentration, and bsp"

Revised as suggested.

Line 345: Was sulfate fully neutralized for the duration of the study?

Sulfate was fully neutralized by $NH_4^+$, $Na^+$, $K^+$ and $Ca^{2+}$ according to the ISORROPIA II model.

Line 355: Was 5% used here?

We used 5.3% as stated in the supplementary. We have added such explanation in the revised paper: "Alternatively, crustal element oxides mass concentration was estimated from $Ca^{2+}$ mass concentration because of their good correlations (slope=0.053, $R^2$=0.79) as was found in a previous study (Fig. S6) (Tao et al., 2017b). It was suggested that $Ca^{2+}$ accounted for 5.3% of crustal element oxides in $PM_{2.5}$ in urban Guangzhou, a value that is close to the content of $Ca^{2+}$ (5.0%) in soil dust source profiles (representing crustal element oxides) in $PM_{2.5}$ in cities of southern China (Sun et al., 2019). Because $CaSO_4$ and $Ca(NO_3)_2$ were mainly from the reactions between calcium oxide and acids (e.g. $H_2SO_4$ and $HNO_3$), the estimated mass concentration of crustal element oxides needs to deduct those of $CaSO_4$ and $Ca(NO_3)_2$."

Line 362: I am not sure what is meant here by the "total"? How was "total" derived in this context?

We have deleted the word "total". We originally referred to the sum of the condensation, droplet and coarse mode mass concentrations.

Line 371: Same comment as the previous.

The word "total" is now deleted.

Line 382: Please provide more details regarding this method.

We have added the formula for calculating number concentration of chemical species in section 2.4, which reads: "$N_{i,j}$ is number concentration of chemical species calculated by the formula (3).

$$N = \frac{6C}{\pi \rho D^3} \tag{3}$$

Where, N is chemical species number concentration, C is chemical species mass concentrations, $\rho$ is density of chemical species (Table S1), D is geometric diameter ($D_g$) of chemical species.

The particle number concentrations in aerodynamic diameter ($D_a$) were converted to

the particle number concentrations in $D_g$ (similar to $D_p$) according to:

$$D_a = D_g/(\rho)^{0.5} \tag{4}$$

$$\rho = \frac{\sum_{chemical\ species} m_i}{\sum_{chemical\ species} \frac{m_i}{\rho_i}} \tag{5}$$

Where, $\rho$ represents the daily average densities of particle, i is chemical species, $m_i$ is chemical species mass concentration in a bin, $\rho_i$ is chemical species density. The seasonal average densities of particle were calculated in Fig. S4."

Line 384: This would be expected because of the diameter-cubed dependence between number and mass.

We agree with this comment and we deleted the irrelevant statement.

Line 386: What is the difference in the definition of the estimated NMAD of the number concentrations of individual species and the NAMD of particle number concentrations? (individual versus particle?). I think an issue here is that the constant 0.21 m value is meaningless.

The sums of the individual species number concentrations were the particle number concentrations. We have revised the text as follows: "The estimated number mean aerodynamic diameters (NMADs) of the number concentrations of individual chemical species mainly distributed in the range of 100-120 nm. The estimated NMADs of particle number concentrations (sum of individual chemical species number concentrations in the same size bin) were close to about 100 nm in the four seasons, which was larger than the NMADs (30-70 nm) of the simultaneously measured particle number concentrations by the SMPS and APS (Fig. 4)."

Line 392: What densities were used?

The densities of the individual chemical species are listed in Table S1.

Table. S1 The refractive indices and densities of chemical species.

| Chemical species | refractive index | density(g cm$^{-3}$) | Chemical species | refractive index | density(g cm$^{-3}$) |
|---|---|---|---|---|---|
| NaCl | 1.54-0i | 2.16 | Ca(NO$_3$)$_2$ | 1.53-0i | 2.50 |
| NaNO$_3$ | 1.59-0i | 2.26 | H$_2$O | 1.33-0i | 1.00 |
| Na$_2$SO$_4$ | 1.48-0i | 2.68 | OM | 1.55-0i | 1.40 |
| (NH$_4$)$_2$SO$_4$ | 1.53-0i | 1.76 | EC | 1.80-0.54i | 1.50 |
| NH$_4$NO$_3$ | 1.55-0i | 1.73 | crustal element oxides | 1.56-0.01i | 2.66 |
| K$_2$SO$_4$ | 1.49-0i | 2.66 | unidentified fraction | 1.58-0.01i | 2.00 |
| CaSO$_4$ | 1.57-0i | 2.61 | | | |

Line 397: The reasoning here isn't clear. The size segregated chemical mass species concentrations should be dry. Unless the authors mean that particle bound water was associated with a gravimetric measurement, the individual species mass do not include water.

The size segregated particle mass concentrations and chemical species mass concentrations were weighted and estimated under a dry condition (temperature = 25°C and relative humidity = 40%). Besides chemical species, water was also resolved in the size segregated samples according to ISORROPIA II model (Fig. 2). In contrast, the particle number concentrations were measured under dry condition (relative humidity < 30%). Moreover, little water was resolved in the size segregated samples according to ISORROPIA II model. To some extent, chemical species likely internally mixed with chemical species in the real world and resulted in the larger diameter of chemical species than the measured ones under dry condition.

Line 418-420: I am unclear as to why scattering efficiencies are being discussed here?

We agree with this comment and have deleted this part.

Line 421-423: This is the first discussion of these design flaws – are the authors referring to the single bin for the condensation mode?

Yes, it was because we cannot get the MMAD in the condensation mode and cannot accurately estimate the number concentration especially those of <100 nm.

Line 426: How much higher?

We have added this text: "On annual average, the estimated particle number concentrations in the range of 430 nm-10 μm based on the size-segregated chemical species mass concentrations were 33±42% higher than those measured by the SMPS and APS."

Line 429: How do the authors know that EC was internally mixed with OM or inorganic salts during this study?

Here we only tried to interpret the possible reasons of the overestimation of the particle number concentrations by the size-segregated chemical species mass concentrations. We have deleted this statement in the revised paper.

Line 433-434: The reasoning here is unclear. What are the estimation errors and models?

This refers to the estimated particle number concentrations by the size-segregated chemical species mass concentrations using the inversion technique and ISORROPIA II model. We have revised the text as follows: "To some extent, the intercepts represent

the measurement errors of SMPS and APS and estimation errors of the inversion technique and ISORROPIA II models.”

Line 448: The authors need to provide more details on how they derived bsp. What refractive indices did they use, how did they calculate them, which number size distributions did they use, etc.

We have added description of several key input parameters of Mie model for estimating $b_{sp}$, which reads: “Daily $b_{sp}$ was estimated using Mie model (in section 2.4) with input parameters including refractive indices, densities and number concentrations in 401 bins of chemical species (NaCl, NaNO$_3$, Na$_2$SO$_4$, (NH$_4$)$_2$SO$_4$, NH$_4$NO$_3$, K$_2$SO$_4$, CaSO$_4$, Ca(NO$_3$)$_2$, H$_2$O, OM, EC, crustal element oxides and unidentified fraction). The refractive indices and densities of above chemical species are summarized in Table S1.”

Line 455: Why “especially the inversion technique method”?

We agree that this statement is a bit confusing and we have deleted the word “especially”.

Line 460: What do the authors mean that OC was underestimated by the OC/EC protocol?

This is because OC of size-segregated samples is defined as OC1 + OC2 + OC3 + OC4 rather than OC1 + OC2 + OC3 + OC4 + OP due to the low OP concentration in each bin.

Line 468, 472: Do the authors mean “inline” data?

We have replaced the word as suggested.

Line 502: What did the authors use for refractive indices for the “unidentified fraction”?

We have added the refractive indices and densities of chemical species in supplementary (Table S1). The refractive index of the unidentified fraction is 1.58-0.01i.

Line 517, 521: I am not sure what is meant by “particle and chemical species”. What is the distinction?

Particle MSE was estimated by sum of $b_{sp}$ from individual chemical species divided by sum of particle mass concentration. MSEs of individual chemical species was estimated by $b_{sp}$ using Mie model according to its particle number in 401 bins, refractive index and density divided by its mass concentration. Thus, we have clarified this part as follows: “Here, only the MSEs of (NH$_4$)$_2$SO$_4$, NH$_4$NO$_3$, OM, EC, crustal element oxides and unidentified fraction in the condensation, droplet, fine (sum of condensation

and droplet), and coarse modes were estimated (Table 3), considering these chemical species accounted for more than 90% of the estimated $b_{sp}$. However, particle MSEs in the condensation, droplet, fine and coarse modes were estimated by sum of $b_{sp}$ from individual chemical species divided by sum of particle mass"

Line 540: This points back to the previous comments as well. Was "particle MSE" estimated by summed bsp from individual species divided by summed particle mass, or was bsp calculated for "particle", which then would require a "particle" refractive index? It would help if the authors provided details for how these things are calculated (see comment for line 448).

See our clarification in the previous comment.

Line 577: Define MMGD

We have redefined the GMMD as geometric mass mean diameters (MMGD) of chemical species (($NH_4$)$_2SO_4$, $NH_4NO_3$ and OM), which was converted from MMAD and its density according to the formula (6) in section 2.4.

Line 626-627: Sentence is unclear.

We have simplified this part as follows: "Different from the approach used for fine particle MSE, the MSEs of ($NH_4$)$_2SO_4$, $NH_4NO_3$ and OM in the droplet mode were determined using measurement-based their mass size distributions prescribed as log-normal size distributions. MSEs of these chemical species strongly depend on their size-distributions, which were defined here as log-normal distributions with three parameters including mass concentration (in the range of 0.43 - 2.1 µm), MMAD and standard deviation ($\sigma$)."

Line 630: How were sigma values calculated?

We have added the key formulas in section 2.4, which included the calculation method of sigma and MMAD.

Line 653: What does "bulk particle" mean?

The bulk particle means the sum mass concentration of the condensation, droplet and coarse modes. We have revised the text as follows: "and particle mass, $NO_3^-$, OC, $Na^+$, $Ca^{2+}$ and $Cl^-$ in both droplet and coarse modes."

Line 670: Sea salt in the IMPROVE formula is assumed to have a mass mean diameter of 2.5 um, so it is assumed to be in the coarse mode with the tail extending into the PM2.5 mode. Figures and Tables: Table 2: Define size range of condensation, droplet and coarse modes. Again, reporting 0.21 um for all condensation mode MMAD is

meaningless. Define "MMAD" in the caption.

We agree with this comment, and we also suspected that sea salt may distribute in the tail of $PM_{2.5}$. However, we cannot find NaCl in $PM_{2.5}$ according to the ISORROPIA II model. In fact, we found a large amount of $Na_2SO_4$ in $PM_{2.5}$, which would be related with aged sea salt. Here, we referred sea salt as NaCl rather than $Na_2SO_4$. We have deleted 0.21 μm for all condensation modes in Table 2. We defined the size ranges in the different modes in Table 2 and defined MMAD in the subtitle.

Table 3: Define size range of condensation and droplet modes. Again, reporting 0.21 um for all condensation mode MMAD is meaningless. Define "MSE" and "MMAD" in the caption. Include wavelength and relative humidity (Dry = ?%) in the caption or subtitle.

We have deleted 0.21 μm for all the condensation modes in Table 3. We have added the size ranges for the different modes in Table 3. Table caption revised as suggested.

Figure 1: Define "PRD" in the caption.

Revised as suggested.

Figure 2: Was CaSO4 and Ca(NO3)2 subtracted out of the soil formula when using Ca to calculate soil? These figures suggest that EC mass size distributions are larger than OM distributions? Are these stacked? If so EC » SO4 but mass concentrations in Table 2 suggests this is not the case. The presentation is somewhat confusing. Keep the y-axis the same for all seasons for easier comparisons.

We have clarified the relationships between the estimated soil mass concentration and calcium salts ($CaSO_4$ and $Ca(NO_3)_2$) in section 3.2.1, which reads: "Alternatively, crustal element oxides mass concentration was estimated from $Ca^{2+}$ mass concentration because of their good correlations (slope=0.053, $R^2$=0.79) as was found in a previous study (Fig. S6) (Tao et al., 2017b). It was suggested that $Ca^{2+}$ accounted for 5.3% of crustal element oxides in $PM_{2.5}$ in urban Guangzhou, a value that is close to the content of $Ca^{2+}$ (5.0%) in soil dust source profiles (representing crustal element oxides) in $PM_{2.5}$ in cities of southern China (Sun et al., 2019). Because $CaSO_4$ and $Ca(NO_3)_2$ were mainly from the reactions between calcium oxide and acids (e.g. $H_2SO_4$ and $HNO_3$), the estimated mass concentration of crustal element oxides needs to deduct those of $CaSO_4$ and $Ca(NO_3)_2$."

The mass size distributions of OM were in fact larger than those of EC in four seasons, although Figure 2 seems to show an opposite result, which was due to the overlap of chemical species. The annual average size distributions of the individual species of $(NH_4)_2SO_4$ OM and EC are plotted below. We have also revised the scale of the y-axis in all the figures.

[Figure]

Fig. Suppl. Continuous log-normal size distributions of $(NH_4)_2SO_4$, OM and EC.

Figure 3: Similar comments as previous caption.

See our response in the previous comment.

Figure 4: It would help to plot the APS data in terms of mass or volume instead of number based on the size range- the larger modes would be more visible.

Although it may be more visible using mass or volume data than using number data, it is the number concentration that was directly measured by SMPS and APS. Moreover, the input data of Mie model also need number concentration of chemical species. Thus, we used the measured number concentrations by SMPS and APS to close the estimated number concentrations of chemical species.

Figure 6: Include wavelength, relative humidity conditions, and size range. Is this total bsp?

We revised the caption as follows: "Fig. 6. Correlations between the measured $b_{sp}$ in TSP at wavelength of 520 nm under dry condition (relative humidity <30%) and estimated $b_{sp}$ in $PM_{10}$ at wavelength of 550 nm under dry condition (relative humidity =40%) in four seasons."

Figure 7: Similar comments to figure 2. I don't understand how the mass of sulfate can be so much higher than EC yet the EC scattering is greater?

Figure 7 shows the contributions of chemical species including $NaCl$, $NaNO_3$, $Na_2SO_4$, $NH_4NO_3$, $(NH_4)_2SO_4$, $K_2SO_4$, $CaSO_4$, $Ca(NO_3)_2$, $H_2O$, OM, EC, crustal element oxides and unidentified fraction to the estimated $b_{sp}$ in the different sizes (0.1-10µm in 401 bins). The different size distributions between $(NH_4)_2SO_4$ and EC caused the higher contribution from $(NH_4)_2SO_4$ to the estimated $b_{sp}$ despite its lower mass concentrations.

[Figure]

Fig. Suppl. Continuous log-normal size distributions of $(NH_4)_2SO_4$, OM and EC.

[Figure]

Fig. S3: Single particle scattering efficiencies of the dominant chemical species.

Figure 8: Define "MSE", "fine", wavelength, and relative humidity conditions in the caption.

Figure caption has been revised as suggested.

Figure 9: Caption doesn't include any information on MSE. Also include wavelength, and relative humidity conditions in the caption.

Figure caption has been revised as suggested.

---

## Author Comment (AC2) · 11 May 2019

**Response to Reviewer #2**

We greatly appreciate the reviewer for providing the detailed comments, which have helped us improve the paper quality significantly. We have addressed all of the comments carefully as detailed below. The original comments are in black and our replies are in blue.

One thing that I recommend doing is to calculate your MSEs also by using multiple linear regression (MLR). Now you calculated them with a Mie model. That is fine and scientifically justified but it also has its uncertainties, for instance related to refractive indices etc. Your data is good for MLR and that would give another estimate for the MSEs. MLR is quick and easy to do – even with Excel – and that is also actually inversely the way air quality data would be used for estimating visibility from PM2.5 filter data. Doing that you would have an additional uncertainty estimate and a closure of MSEs.

We agree with the reviewer that it is worth and relatively easy to use the multiple linear regression (MLR) model to estimate MSEs, as we have done in several of our previous studies (Tao et al., 2014a, 2014b, 2015, 2016). In fact, we have recently completed another study comparing MSEs calculated from using various methods including the MLR model. We chose not to present the results here from this particularly method because (1) the paper is already very long, (2) the focus of the present study is to investigate the causes of the variations in the estimated MSEs (not the absolute errors in using the Mie model), and (3) a systematic study on model differences in the calculated MSEs will be presented in a separate study.

Having done that I suggest you make an additional scatter plot and linear regressions of scattering coefficient calculated with the Mie-derived MSEs, with the MLR-derived MSEs and with IMPROVE MSEs vs. measured scattering coefficient. Now you have written in the text new MSEs and written how they differ from the IMPROVE MSEs but the full comparison for the Guangzhou air is missing, that would be the linear regressions I suggested. How well do the different MSEs predict the observed scattering?

As explained in the previous comment, we chose not to present the MLR-derived MSEs in this study. Here, we focused on comparing the differences in the estimated $b_{sp}$ using the estimated MSEs of chemical species and the measured $b_{sp}$ in section 3.3.1. We have revised the explanation as follows: "Generally, good correlations ($R^2 > 0.79$) were found between the measured and estimated $b_{sp}$ using the average MSEs of chemical species in Table 3 with the slopes being 0.85, 0.84, 0.76 and 0.84 in spring, summer, autumn and winter, respectively (Fig. S8). Thus, the estimated MSEs of chemical species in Table 3 were underestimated."

[Figure]

Fig. S8: Correlations between the measured $b_{sp}$ (<100 μm) at wavelength of 520 nm under dry condition (relative humidity <30%) and estimated $b_{sp}$ (<10 μm) using average MSEs of chemical species at wavelength of 550 nm under dry condition (relative humidity =40%) in four seasons.

The estimated $b_{sp}$ depended on both the mass concentrations and MSEs of chemical species. Thus, it is difficult to assess the difference in the estimated $b_{sp}$ only using the MSEs chemical species. In fact, there are large uncertainties from mass concentrations. As mention above, we have recently finished another study to address this issue, which tentatively titled "The differences in the estimated particle scattering coefficient using the different methods in urban Guangzhou of South China".

As Fig 1 and 2 show, on annual average, the estimated mass concentrations of $(NH_4)_2SO_4$ and $NH_4NO_3$ in $PM_{2.5}$ using the ISORROPIA II model were 42±24% and 33±44%, respectively, lower than those using the original IMPROVE formula. However, the estimated MSEs of the $(NH_4)_2SO_4$, $NH_4NO_3$ and OM in the fine mode using the multiple models were 47%, 50% and 15%, respectively, higher than those in the original IMPROVE formula. As a result, the differences in annual average contributions of the dominant chemical species were less than 3% between using the multiple models and the original IMPROVE formula. In contrast, the estimated mass concentrations of $(NH_4)_2SO_4$, $NH_4NO_3$ and organic matter (OM) using the multiple models were 93±16%, 96±9% and 60±32%, respectively, lower in the small mode and 20±50%, 674±569% and 43±68%, respectively, lower in the large mode than those from using the revised IMPROVE formula. The differences in the estimated MSEs of $(NH_4)_2SO_4$, $NH_4NO_3$ and OM were less than 13% between using the multiple models and the revised IMPROVE formula. Generally, the estimated contributions of the dominant chemical species ($(NH_4)_2SO_4$, $NH_4NO_3$ and OM) to the measured $b_{sp}$ under

dry condition using the original and revised IMPROVE formula were acceptable.

[Figure]

Fig. Suppl.1. Correlations between the measured $b_{sp}$ and the estimated $b_{sp}$ using the multiple models (a), the original IMPROVE formula (b) and the revised IMPROVE formula (c).

[Figure]

Fig. Suppl.2. Contributions of chemical species to the measured $b_{sp}$ using the multiple models, the original IMPROVE formula and the revised IMPROVE formula.

Another thing I miss is equations. For example equations of how you calculated MSE, the mean diameters you are using and also chemistry: Did you dry the sampling air for the impactor? If not the particles are larger and get collected on the upper stages which affects the inverted size distributions and ultimately the Mie-modeled scattering. At least some discussion of this would be good.

We have added key formulas in section 2.4. We have revised the text as follows: "Particle MSE was estimated by the sum of $b_{sp}$ from individual chemical species divided by sum of particle mass concentration according to:

$$\text{MSE} = \frac{\int_0^{D_{i,j}^{max}} b_{sp}\ dD_{i,j}}{\int_0^{D_{i,j}^{max}} C\ dD_{i,j}} \tag{1}$$

Where i is chemical species, j is chemical species size, $D_{i,j}$ is the chemical species diameter, and C is chemical species mass concentration."

Yes, the size segregated samples were collected under the ambient condition rather than the dry condition. We believe the MMADs of chemical species under the ambient would be larger than those under the dry condition due to the particle hygroscopic properties. However, we cannot quantify the difference in the size distribution under the ambient and dry conditions. We highlighted this factor in the analysis of closure between the measured and estimated $b_{sp}$ in section 3.2.3. We have revised the discuss as follows: "Moreover, the size distributions would be different under dry and ambient conditions due to the particle hygroscopic properties. In fact, the NMADs of particle measured by SMPS and APS under dry condition were less than those measured by the size-segregated sampler under ambient condition according to section 3.2.2. Thus, the estimated $b_{sp}$ based on size distributions of chemical species would be systematically higher to some extent than the measured $b_{sp}$ under dry condition."

Detailed comments
L131 " ... geometric diameter (Dg) ..." The widely used meaning of Dg is the geometric mean diameter of a particle number size distribution. So use Dp. for the aerodynamic diameter use Da.

We have revised the descriptions as follows: "Particle number concentration for particles in the range of 14 nm - 615 nm in mobility diameter ($D_p$) was measured….."

L137-138. Nephelometer: did you calibrate it?

We have added the statement: "Zero calibration was performed every day with zero air, and span check was done every 3 days using HFC-R134a gas."

L197: explain the Mie model in a bit more detail.

We have added the following description in section 2.4:

$b_{sp}$ was estimated by the Mie model as follows:

$$b_{sp} = \int_0^{D_{i,j}^{max}} \frac{\pi}{4} D_{i,j}^2 Q_{sp}(m_{i,j}, D_{i,j}, \lambda) N_{i,j} dD_{ij} \tag{2}$$

Where $Q_{sp}$ is single-particle scattering efficiency of chemical species (Fig. S3), $m_{i,j}$ is refractive index of chemical species (Table S1), $\lambda$ is 550 nm, and $N_{i,j}$ is number concentration of chemical species calculated by the formula (3).

$$N = \frac{6C}{\pi \rho D^3} \tag{3}$$

Where N is chemical species number concentration, C is chemical species mass concentrations, $\rho$ is density of chemical species (Table S1), and D is geometric diameter ($D_g$) of chemical species.

The particle number concentration in aerodynamic diameter ($D_a$) was converted to the particle number concentration in $D_g$ (similar to $D_p$) according to:

$$D_a = D_g/(\rho)^{0.5} \tag{4}$$

$$\rho = \frac{\sum_{chemical\ species} m_i}{\sum_{chemical\ species} \frac{m_i}{\rho_i}} \tag{5}$$

Where $\rho$ represents the daily average density of particle, $i$ is chemical species, $m_i$ is chemical species mass concentration in a bin, and $\rho_i$ is chemical species density. The seasonal average densities of particle are shown in Fig. S4.

L201, define MMAD and give the formula L206 "limit of detection" is wrong here, that expression is related to concentration measurements

We have given the formula for calculating MMAD in section 2.4. We believe MMAD was related with the mass concentrations in each bin of size-segregated sampler. However, only one bin is designed in the condensation mode, and we thus used formula (9) to estimate MMAD in the condensation mode.

L248 "As expected" – why would you expect this?

It was because $SO_4^{2-}$, $NO_3^-$ and $NH_4^+$ are mainly formed through aqueous-phase reactions in moisture conditions in the PRD region. Thus, most of them should be distributed in the droplet mode.

L266 "NO3 mainly exists in the form of ammonium nitrate..." you have data on the inorganic ion concentrations but how did you calculate concentration of ammonium sulfate and ammonium nitrate? Give a couple of formulas.

The chemical species including $NH_4NO_3$, $(NH_4)_2SO_4$ were estimated by the ISORROPIA II model, which was run at the reserved mode with input data of $K^+$, $Ca^{2+}$, $Mg^{2+}$, $NH_4^+$, $Na^+$, $SO_4^{2-}$, $NO_3^-$, $Cl^-$, RH (40%), and temperature (25°C). The ISORROPIA II model has an open source code. The key formulas were described in Fountoukis and Nenes (2007).

L385 "NMAD" – give formula

The NMAD is the number mean aerodynamic diameter, which is calculated the same way as the MMAD (mass mean aerodynamic diameter) except of substituting the mass concentration with number concentration. The actual formula for NMAD calculation can be expressed as:

$$NMAD = \frac{\int_0^{D_i^{max}} \frac{6C}{\pi \rho D_i^2}}{\int_0^{D_i^{max}} \frac{6C}{\pi \rho D_i^3}}$$

L574 "mass median geometric diameter (MMGD)" I have never heard of. Define. Consider using some other descriptive diameter that has been presented in literature.

The mass median geometric diameter derived from mass median aerodynamic diameter. The definition of the geometric mass mean (or median) diameters and mass mean (or median) aerodynamic diameters can be found in Hand and Malm (2007). Thus, we revised mass median geometric diameter (MMGD) as geometric mass mean diameters (GMMD).

Hand, J. L., and Malm, W. C.: Review of the IMPROVE equation for estimating ambient light extinction coefficients, CIRA, Colorado State University, 2007.

Fig. 4. Are the diameters of the SMPS data and the APS data both aerodynamic or what? The gap is huge, try to explain it.

The diameter of the SMPS data is $D_p$, while the diameter of the APS data is $D_a$. the gap of number concentrations between using SMPS and using APS were mainly due to the different dlog(D). The dlog($D_p$) and dlog($D_a$) were 0.015 and 0.031, which meant the measured number concentration by SMPS would be higher by 2 times (0.031/0.015) than those measured by APS at the same size.

Fig 5. The numbers in the x and y axes cannot be true. In Guangzhou number concentrations are in the range of thousands, now the max concentration is about 400 /cc.

Total particle number concentration in the range of 10 nm-10 μm measured by the SMPS and APS were 7038±2250 $cm^{-3}$, 9774±1471 $cm^{-3}$, 5694±1942 $cm^{-3}$ and 10801±2986 $cm^{-3}$, respectively, in spring, summer, autumn and winter. As shown in Fig. 4, most of particles distributed in the condensation mode (<430 nm). Here, Fig. 5 shows the correlations between the estimated and SMPS- and APS-measured particle number concentrations (430 nm-10 μm) in four seasons. To avoid misunderstanding, we revised the figure caption as follows: "Fig. 5. Correlations between the estimated and SMPS- and APS-measured particle number concentrations in the size range of 430 nm-10 μm in four seasons."